# LEARN ALL YOU NEED IN ONE HYPERNETWORK

## ABSTRACT

While the attention mechanism is considered the cornerstone of Transformers, its layer-specific parameterization presents challenges for efficiency and knowledge reuse. Recent work reformulates multi-head self-attention as a hypernetwork, suggesting it can be mathematically interpreted as an implicit hypernetwork conditioned on key–query pairs. However, prior work has been limited to small-scale tasks or theoretical demonstrations, leaving open whether explicit hypernetworks can scale to full language-model pre-training. We first prove the existence of a shared hypernetwork that can approximate the multi-head self-attention with fewer parameters. Building on this insight, we propose *HyperBERT*, a BERT-style Transformer encoder in which the multi-head self-attention mechanism is replaced by a single-layer MLP dynamically generated by one explicit, shared hypernetwork. In our experiments, a 4-head, 2-layer Transformer decoder serves as the shared hypernetwork to generate a single-layer MLP to replace all query, key, value, and output (QKVO) projection matrices in each layer of a 4-head, 4-layer BERT. Pre-trained on WikiText-103, our 4-layer HyperBERT matches the average GLUE score of a BERT baseline ($\Delta \leq 0.1$) with 6% fewer parameters and outperforms other MLP-based attention alternatives. Furthermore, the transplant experiment shows that the hypernetwork's learned weights transfer more effectively to deeper models than conventional attention parameters under a fixed-parameter budget. To the best of our knowledge, this is the first pre-training study that replaces multi-head self-attention with MLPs generated by a shared hypernetwork. Our results suggest that an explicit, shared hypernetwork can serve as a modular, parameter-efficient replacement for multi-head self-attention in BERT-style Transformer encoder models while preserving language modeling capabilities.

## 1 INTRODUCTION

Transformers (Vaswani et al., 2017) built on multi-head attention have revolutionized NLP and beyond. However, their scalability comes at the cost of enormous parameter counts and quadratic computational complexity. Each Transformer layer carries its own set of query, key, value, and output matrices for every attention head, leading to a proliferation of parameters that are not shared across layers. This duplication limits cross-layer reuse of computation, thereby limiting the modularity of the Transformer's components.

Recent studies (Behnke & Heafield, 2020; Shim et al., 2021; Michel et al., 2019) have shown that attention maps are highly similar and not all heads contribute significantly to performance. A large percentage of attention heads can be removed with minimal impact on accuracy. This has motivated studies in modular architectures that simplify the roles of attention (Tay et al.; Liu et al., 2021; Tolstikhin et al., 2021). A compelling idea (Mai et al., 2023) to reduce computational complexity is to dynamically generate layer-specific parameters to replace quadratic multi-head attention computation with separate neural networks (i.e., HyperNetworks (Ha et al., 2017)). Moreover, recent studies (Schug et al., 2025; Schlag et al.) have shown that multi-head attention can be viewed implicitly as a form of hypernetwork or "fast weight programmer" that constructs context-specific transformations on the fly. A low-dimensional latent code dynamically determines how information is routed. This suggests an intriguing opportunity to explore whether the Transformer's attention mechanism can be externalized to an explicit hypernetwork. By decoupling attention into a separate, trainable parameter-generating module, we can achieve better modularity, where a distinct component learns to "program" layer computations. This could also improve parameter-sharing

across depth and enhance the reusability of learned operations. However, prior work (Schug et al., 2025; Mai et al., 2023) has been limited to toy-scale experiments or partial attention modifications.

In this work, we intend to investigate the feasibility of replacing attention matrices with MLPs generated by an explicit, shared hypernetwork, thereby reducing overall model size while preserving both performance and transferability. We propose *Hyper-BERT*, a BERT (Devlin et al., 2019)-style Transformer architecture that replaces multi-head self-attentions with dynamically generated MLP layers produced by a context-conditioned hypernetwork. HyperBERT retains the familiar Transformer encoder structure but replaces the multi-head self-attention sub-layer, i.e., Q, K, V, O projection matrices, with a single-layer MLP whose parameters are generated on the fly by a shared hypernetwork across layers. Concretely, the hypernetwork is a Transformer decoder that reads the entire input sequence as context and outputs customized parameters for the target MLP in each encoder layer. This design means that a single hypernetwork is responsible for all cross-token computa-

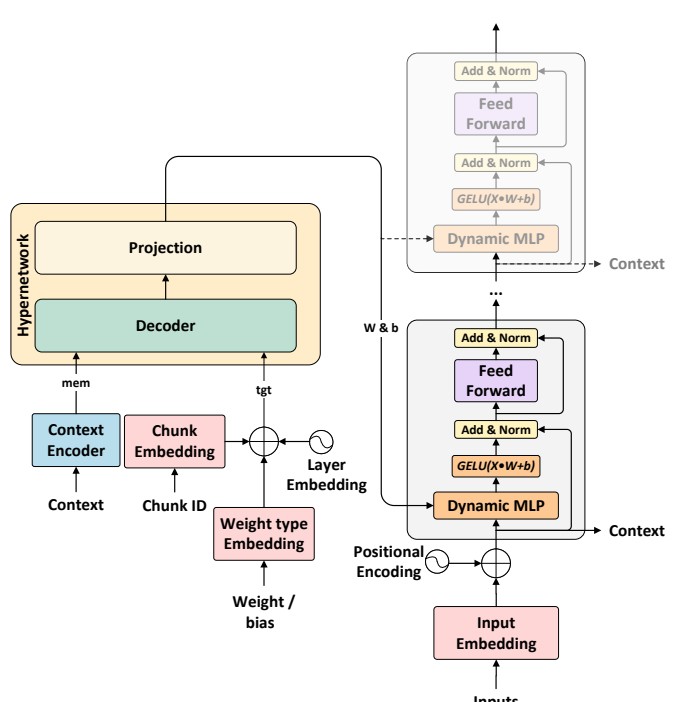

Figure 1: Overview of the HyperBERT Architecture. A shared hypernetwork (left) uses input context and embeddings to dynamically generate a single-layer MLP, which replaces the multi-head self-attention in each encoder layer.

tions across all encoder layers. By decoupling the implicit parameter generation mechanism in multi-head self-attention from the encoder layers, we eliminate separate attention parameters at every layer and reuse one modular mechanism throughout. Notably, the encoder itself is attention-free in its main forward path, yet the hypernetwork still uses attention internally to encode the sequence when generating the MLP's parameters. This explicit hypernetwork architecture is highly modular and generative: a unified Transformer module produces layer-specific computations on demand, which improves parameter efficiency. Figure 1 provides an overview of the HyperBERT architecture.

To assess the impact of removing explicit attention, we pre-train our HyperBERT from scratch on WikiText-103 (Merity et al., 2017) and evaluate it on the GLUE (Wang et al.) benchmark. In our experiments, HyperBERT uses a 4-head, 2-layer Transformer decoder as the shared hypernetwork generating MLP replacements for a 4-head, 4-layer BERT baseline model. Remarkably, we find that HyperBERT matches the performance of the baseline BERT model while using about 6% fewer parameters. Despite removing attention from the encoder layers and using fewer attentions, HyperBERT achieves the same level of language understanding as the BERT baseline and outperforms other MLP-based attention alternatives. To evaluate the transfer learning ability of the learned hypernetwork, we conduct a transplant experiment in which the trained hypernetwork is inserted into a deeper 6-layer model. We then compare its performance with a baseline model with the same number of attention parameters transplanted from the BERT baseline. By reusing the trained hypernetwork, this new 6-layer HyperBERT model outperforms the baseline on the GLUE benchmark.

Beyond empirical results, our work offers a novel theoretical insight: an explicit hypernetwork can replicate the functional role of multi-head self-attention using simpler components. This suggests that multi-head self-attention's power arises from a form of dynamic parameter generation. By learning

multi-head self-attention in one shared hypernetwork, complex attention operations can be factored into independent, reusable modules.

Our main contributions can be summarized as follows.

- We propose HyperBERT, a BERT-style Transformer encoder variant in which multi-head self-attention is entirely replaced by MLPs generated by an explicit, shared hypernetwork. To the best of our knowledge, this is the first study to externalize the implicit weight generation mechanism of multi-head self-attention into a dedicated module and to evaluate the design in a pre-training scenario.

- To ground our approach, we provide a theoretical analysis of the parameter-efficient shared hypernetwork capable of replicating the transformations of all attention layers within a bounded error under the low-rank assumption.

- Through extensive experiments, we show that HyperBERT matches the performance of a BERT baseline on the GLUE benchmark while using 6% fewer parameters. We also show that HyperBERT outperforms other state-of-the-art MLP-based attention alternatives.

- To investigate HyperBERT's transferability, we transplant its trained hypernetwork module into a deeper model. Experiments show that the transplanted HyperBERT outperforms the deeper BERT baseline while using the same number of transplanted parameters.

The paper is structured as follows: Section 2 introduces related work. Section 3 provides a theoretical analysis. Section 4 describes HyperBERT. Section 5 shows the experimental results. Finally, Section 6 concludes the paper.

## 2 RELATED WORK

**Redundancy and pruning in multi-head attention.** Multi-head attention often exhibits significant redundancy, with many heads contributing little unique information. Recent studies (Michel et al., 2019; Behnke & Heafield, 2020; Shim et al., 2021; Dong et al.; Anagnostidis et al., 2022; Amsel et al., 2025; Bhojanapalli et al.; Jin et al., 2024) have shown that a large percentage of attention heads can be removed after training without significantly impacting the performance. In fact, only a subset of heads perform the "heavy lifting", while the rest can be pruned with minimal impact. The last heads to be removed tend to be those with linguistically interpretable roles. These findings suggest that the expressiveness gained by multiple heads is not fully utilized, motivating simpler approaches to replace the attention mechanism while preserving performance.

**Hypernetworks in Transformers and NLP.** Hypernetworks (Ha et al., 2017) are neural networks that generate the parameters of other neural networks. Ha *et al.* (Ha et al., 2017) first demonstrated that a small hypernetwork could produce the weights of a larger network. In the context of NLP and Transformers, hypernetworks have been explored as a way to introduce dynamic adaptability in many tasks such as fine-tuning, multi-task learning, and continual learning (Charakorn et al., 2024; von Oswald et al., 2020; Karimi Mahabadi et al.; Navon et al., 2021; Zhao et al.). Recent studies (Karimi Mahabadi et al.; Charakorn et al., 2024) use hypernetworks to generate task-specific adapter layers for Transformer blocks. This allows a single model to condition on a task embedding and output customized adapter parameters for different tasks. Furthermore, there are studies (von Oswald et al., 2020) that use hypernetworks to reduce catastrophic forgetting in continual learning. Beyond these, hypernetworks have shown their efficacy in domains such as neural architecture search (Zhang et al., 2019; Knyazev et al., 2021; Knyazev et al.).

**Alternatives to attention mechanisms.** The limitations of standard attention have inspired many alternative architectures that modify or replace the attention module. Synthesizer (Tay et al.) removes the token-token dot product interaction entirely. Instead, Synthesizer learns synthetic attention weight matrices that are independent of pairwise query-key similarity. gMLP (Liu et al., 2021) is an architecture using MLPs with gating that enables token interaction without any explicit attention mechanism. Building on the success of MLP-based models, MLP-Mixer (Tolstikhin et al., 2021) and its variants propose to replace attention with fixed-weight token mixing MLPs. However, a static MLP mixer has limited capacity to adapt to varying inputs. HyperMixer (Mai et al., 2023) addresses this by generating the token-mixing MLP dynamically via a hypernetwork. Instead of a fixed mixing pattern, HyperMixer's hypernetwork generates dynamic MLP weights, allowing flexible token interaction with inductive biases similar to Transformers. In contrast to prior approaches,

HyperBERT uses the same hypernetwork shared across all layers and replaces all attention heads with generated MLPs. Furthermore, HyperBERT scales to the pre-training scenario to fully explore its language modeling ability.

**Theoretical perspectives: attention as weight generation.** Recent theoretical work has reinterpreted the attention operation itself as a form of implicit fast weight generator or hypernetwork. (Schug et al., 2025; Schlag et al.; Galanti & Wolf, 2020; Tong et al., 2025) Schlag et al. demonstrated a formal equivalence between linear Transformers and the early fast weight programmer models from the early 1990s. In linear Transformers, the update of the attention output can be seen as a slow neural net writing to a fast-changing weight matrix. This connects the attention mechanism with the idea of one network dynamically programming the weights of another. More recently, Schug et al. (2025) reformulated standard multi-head attention as an implicit hypernetwork. They show that a low-dimensional latent code derived from each query-key pair effectively parameterizes the linear transformation applied to the values. These perspectives reinforce the view that the power of attention lies in its ability to produce context-conditioned operations.

## 3 THEORETICAL ANALYSIS

In this section, we present a theoretical analysis that builds on prior theoretical and empirical results (Schug et al., 2025; Schlag et al.; Michel et al., 2019; Behnke & Heafield, 2020; Shim et al., 2021; Dong et al.; Anagnostidis et al., 2022; Amsel et al., 2025; Bhojanapalli et al.; Jin et al., 2024; Naderi et al., 2025; Galanti & Wolf, 2020; Yun et al., 2020). Our goal is to show that there exists a shared hypernetwork that can generate transformations that replicate self-attention perfectly. Furthermore, under low-rank assumptions, we argue that a more compact shared hypernetwork can approximate the Transformer attention mechanism well.

### 3.1 NOTATION AND PROBLEM STATEMENT

- Let $T \in \mathbb{N}$ be the maximum sequence length and $d_{model} \in \mathbb{N}$ the embedding dimension.
- For every layer $l \in \{1, \ldots, L\}$ a standard attention block is parameterized by a vector:

$$w_l \in \mathbb{R}^P \text{ where } P = 4d_{model}^2 \text{ for query, key, value, and output projections} \tag{1}$$

Its forward map is:

$$f_l : \mathbb{R}^{T \times d_{model}} \longrightarrow \mathbb{R}^{T \times d_{model}}, \qquad x \mapsto F(x; w_l). \tag{2}$$

where $F$ is the usual attention equation. Suppose

$$\sup_{x \in \mathcal{X}} ||F(x; w_l) - g_l(x)||_2 \leq \epsilon \tag{3}$$

for some ground-truth function $g_l$ and tolerance $\epsilon > 0$. The total parameter budget of the baseline is therefore $LP$.

### 3.2 UPPER-BOUND CONSTRUCTION: DUMMY SHARED HYPERNETWORK

Define an embedding $e_l \in \mathbb{R}^L$ that is one-hot at position $l$. A neural network $H_{dummy} : R^L \to R^P$ with weights

$$\Theta_{dummy} = [w_1^\intercal \ldots w_L^\intercal]^\intercal \in \mathbb{R}^{L \times P} \tag{4}$$

acts as a table lookup: $H_{dummy}(e_l) = w_l$.

Its parameter count equals $LP$ and it reproduces every $f_l$ with error $\epsilon$. This is an existence proof that some shared hypernetwork across layers can match every layer's $f_l$. Our goal is to shrink its size.

### 3.3 LOW-RANK STRUCTURE OF LEARNED ATTENTION WEIGHTS

Previous studies (Michel et al., 2019; Behnke & Heafield, 2020; Shim et al., 2021; Dong et al.; Anagnostidis et al., 2022; Amsel et al., 2025; Bhojanapalli et al.; Jin et al., 2024) report strong redundancy across heads and layers.

**Assumption 1** (Low-Rank Assumption). *Formally, let $W$ be the stack of $w_l$*

$$W = [w_1 \ldots w_L] \in \mathbb{R}^{P \times L}. \tag{5}$$

*There exists an integer $r \leq min\{P,L\}$ and matrices $U \in \mathbb{R}^{P \times r}, V \in \mathbb{R}^{r \times L}$ such that*

$$rank(W) = r, W = UV \tag{6}$$

In practice, weight matrices of trained neural networks are rarely low-rank in a strict mathematical sense. Therefore, to empirically validate the practical implications of this assumption, we instead analyze the effective rank of $W$. The effective rank is the number of singular values required to preserve a percentage of the matrix's cumulative energy. We analyze 17 diverse pre-trained Transformer models with depths from 6 to 32 layers. We compute the effective rank of the stacked attention matrices required to preserve 80%, 85%, and 90% of the cumulative energy. We then modeled the relationships with a power law $r@\text{Threshold} = \alpha \cdot L^\beta$, where $\beta < 1$ indicates sublinear growth. The provided results suggest that the low-rank assumption is practically valid for a wide range of modern Transformer models. Our analysis shows that the effective rank of stacked attention weight matrices grows sublinearly with model depth (See Appendix C).

### 3.4 Constructing a compact hypernetwork

Define two components

- **Basis memory** $U$ that is shared and trainable with the size $Pr$.
- **Coefficient generator** $G_\phi : R^m \to R^r$ with parameter $\phi$, where the input is a layer code $c_l$. $c_l$ can be an embedding of $l$ or a learned summary of the current hidden state.

Then we can construct a shared hypernetwork as

$$H_\theta(c_l) = UG_\phi(c_l), \theta = (U, \phi). \tag{7}$$

Its output dimension is P, so it can replace $w_l$ inside F.

### 3.5 Approximation theorem

**Theorem 1.** *Assume that (1) $\text{rank}(W) = r$ with $r < L$ and (2) the coefficient generator $G_\phi$ is a feed-forward network with a single hidden layer of width $2r + 1$ and ReLU activations.*

*Then there exists parameters $\theta^*$ such that*

$$\sup_l \sup_{x \in \mathbb{X}} ||F(x; H_{\theta^*}(c_l)) - g_l(x)||_2 \leq K\epsilon \tag{8}$$

*while the total number of trainable parameters satisfies*

$$\underbrace{Pr}_{U} + \underbrace{(2r + 1)(m + r)}_{\phi} < LP \quad \text{whenever} \quad r + \underbrace{\frac{(2r + 1)(m + r)}{P}}_{\approx 0} < L \tag{9}$$

**Proof sketch of Theorem 1** Factorization gives $w_l = Uv_l$ with $v_l \in \mathbb{R}^r$. Since the index set $\{v_l\}_{l=1}^L$ is finite, the universal approximation theorem (Kim et al.) ensures that a width $2r + 1$ ReLU network $G_\phi$ can approximate the mapping $c_l \mapsto v_l$ to arbitrary precision, hence $UG_\phi(c_l)$ approximates $w_l$. Replacing the $w_l$ by the hypernetwork output in $F$ increases the approximation error by at most $K\epsilon$.

### 3.6 Discussion

The dominant term in the total number of trainable parameters of the shared hypernetwork is $Pr$. As soon as the empirical rank r grows sublinearly in L, the shared hypernetwork has a smaller size than $PL$. The construction of parameter-efficient shared hypernetworks is compatible with conditioning on hidden states instead of a layer index. Consider a hypernetwork conditioned on a hidden state with the size of $d_{model}$, the total number of trainable parameters in the hypernetwork becomes $\underbrace{Pr}_{U} + \underbrace{(2r + 1)(\frac{\sqrt{P}}{2} + r)}_{\phi}$. To ensure $\underbrace{Pr}_{U} + \underbrace{(2r + 1)(\frac{\sqrt{P}}{2} + r)}_{\phi} < LP$, we need that $r + \frac{(2r+1)(\frac{\sqrt{P}}{2}+r)}{P} < L$. Since $P$ is the dominant term, we can claim that $L_{min} \approx r + 1$.

We have shown that there exists a shared parameter-efficient hypernetwork that can preserve the language modeling ability of the vanilla Transformer encoder under the Assumption 1. Given that the effective rank grows sublinearly with the model depth, we further prove that the parameter efficiency benefit can be scaled to deeper models. The proof is provided in Appendix D. We have also shown how to construct such a shared hypernetwork via low-rank decomposition. However, it still requires

trained attention layers. Therefore, we propose an end-to-end trainable HyperBERT that directly learns the shared hypernetwork parameters.

For computational efficiency, Appendix D provides the full per-layer floating-point operations under standard Transformer conventions. In particular, we obtain the closed-form threshold in the sequence length $T$ where the BERT's quadratic term $2T^2d$ exceeds the hypernetwork overhead, and visualize this crossover across $(d, h)$ settings (Fig. 2).

# 4 HYPERBERT

HyperBERT is a Transformer (Vaswani et al., 2017) encoder architecture modeled on BERT (Devlin et al., 2019) with a change in its core attention mechanism. Instead of multi-head self-attention, each layer uses a dynamically generated single-layer MLP as the first sub-layer. The parameters of this MLP are not fixed and are generated on the fly by a separate hypernetwork (Ha et al., 2017) module shared across all layers. The remainder of the Transformer layer remains identical to the standard BERT. This design is motivated by the goal of introducing context-dependent adaptability and parameter efficiency. By generating attention-replacement parameters conditioned on the input sequence, HyperBERT can tailor its transformations to each sequence's content as attention does.

## 4.1 HYPERNETWORK ARCHITECTURE FOR WEIGHT GENERATION

**Hypernetworks.** At the heart of HyperBERT is a hypernetwork that generates the weight matrix and bias vector for the MLP sub-layer in each encoder layer. A hypernetwork is a neural network $h(x; \theta)$ that can generate parameters of other neural networks conditioned on some input $x$. We denote this mapping as $(W, b) = h(x; \theta)$ where $W$ is the generated weight matrix and $b$ is the bias vector. The hypernetwork can be an arbitrary neural network, including a simple MLP. In our implementation of HyperBERT, the hypernetwork is implemented as a small Transformer decoder consisting of two decoder layers, and each decoder layer uses a 4-head cross-attention. This module is shared across all encoder layers, meaning that the same hypernetwork generates parameters for every layer's MLP sub-layer.

**Context encoding.** Each encoder layer's input in the original BERT architecture is provided as context to the hypernetwork. Instead of compressing it into a low-dimensional latent code, we feed the full input sequence $x \in \mathbb{R}^{T \times d_{model}}$ with max length $T$ and embedding dimension $d_{model}$ into the hypernetwork. To prepare the context, we apply a simple two-layer MLP with a GELU activation in between to each token's representation. This context encoder projects the token features from embedding dimension $d_{model}$ to the context dimension $d_c$. This yields an encoded context sequence $C = [c_1, c_2, \ldots, c_T] \in \mathbb{R}^{T \times d_c}$. The context dimension $d_c$ is a hypernetwork-specific model size. By retaining the full sequence rather than a summary vector, the hypernetwork can attend to specific tokens and patterns in the input when generating the parameters.

**Embeddings.** To reduce the size of the hypernetwork's output space, the hypernetwork generates the weight matrix in a row-wise fashion using learnable chunk embeddings. We allocate one chunk embedding for each row of the weight and each element of the bias vector. Formally, if the model dimension is $d_{model}$, we maintain $d_{model}$ learnable chunk embeddings, each corresponding to a row index $1, \ldots, d_{model}$. In addition, we use a small type embedding to inform the hypernetwork of the type of parameter being generated. The weight matrix generation uses a type ID 0, and the bias generation uses type ID 1, with a learned embedding vector for each. Finally, a layer embedding is used to inform the hypernetwork of the type of layer when layer-specific conditioning is used. The chunk embedding, type embedding, and layer embedding are summed and then projected via a linear layer into the context dimension $d_c$. This produces an embedding chunk with the size of $\mathbb{R}^{d_{model} \times d_c}$, which will be used as the target sequence for decoder layers.

**Transformer decoder for row-wise decoding.** The hypernetwork's Transformer decoder takes the projected embeddings with the size of $\mathbb{R}^{d_{model} \times d_c}$ as its target sequence and the encoded context with the size of $\mathbb{R}^{T \times d_c}$ as its source memory. We do not apply causal masking to the decoder. The decoder operates for a fixed, small number of layers (two layers in our design). After decoder layers, we obtain a final state for each query with length $d_c$. These states are then passed through output heads to produce the actual weight and bias values. For the weight matrix, the decoder's output is projected to a $d_{model}$-dimensional vector. Collecting all $d_{model}$ such vectors yields a generated weight matrix of shape $[d_{model}, d_{model}]$. For the bias vector, we perform a similar process, but add the bias-type embedding. The generated $d_{model}$ output vectors are each projected down to form elements of the

bias. This results in a bias vector of shape $[d_{model}]$. The result is that the hypernetwork outputs a complete set of parameters for an MLP: $W \in \mathbb{R}^{d_{\text{model}} \times d_{\text{model}}}$ and $b \in \mathbb{R}^{d_{\text{model}}}$. Importantly, the same hypernetwork is used at every layer and for every input. It learns a general mapping from input contexts to appropriate parameters and acts as a universal parameter generator for the model.

### 4.2 ATTENTION-REPLACEMENT MLP SUB-LAYER

Once the hypernetwork yields the weight matrix $W$ and bias $b$ for a given layer and corresponding input, these parameters are used to perform the actual transformation in place of multi-head self-attention. The generated weight matrix is applied as a standard linear projection across $d_{model}$, and we incorporate a non-linearity to form a one-layer MLP. Concretely, if $X \in \mathbb{R}^{T \times d_{\text{model}}}$ is the sequence of input embeddings to the layer with sequence length $T$, we compute $Y = \text{GELU}(X \cdot W + b)$ where $W \in \mathbb{R}^{d_{\text{model}} \times d_{\text{model}}}$ and $b \in \mathbb{R}^{d_{\text{model}}}$ are the context-conditioned weight and bias generated by the hypernetwork. This operation is applied position-wise: each token's $d_{\text{model}}$-dimensional representation is transformed by the same MLP. After the linear transform and activation, the output $Y$ has the same shape as the input $X$, allowing it to be combined with the input via a residual connection. Notably, since $W$ and $b$ are conditioned to the entire sequence's context, this transformation $Y = \text{GELU}(XW + b)$ is not static. It is specialized to the particular input sequence $X$. This design allows the model to emulate the effects of attention using a simpler MLP computation at run-time. We provide a theoretical analysis in Appendix D to support our use of one-layer MLPs as a replacement for multi-head self-attention.

### 4.3 INTEGRATION INTO THE TRANSFORMER ENCODER LAYER

The overall encoder layer structure in HyperBERT mirrors that of BERT with the above substitution. Each encoder layer contains two main sub-components: (1) the Hypernetwork-generated MLP sub-layer replacing multi-head self-attention and (2) the standard feed-forward network (FFN) sub-layer. The input sequence $x_i$ to the $i$-th block is first passed into the shared hypernetwork to generate the corresponding MLP in which $W^i, b^i = h^i(x_i; \theta)$. The input $x_i$ is then fed into the dynamic MLP sub-layer with a GELU activation function. After a residual connection and a layer normalization, the result is the output of sub-layer 1, denoted as $H^i(x_i)$. This is then fed into sub-layer 2, which is a conventional Position-wise FFN exactly as in BERT. The FFN takes $H^i(x; \theta)$ as input, produces an output $F^i(x) = \text{FFN}(H^i(x))$, and again a residual connection is added as $H^i(x) + F^i(\text{x})$, followed by layer normalization to yield the final output $Y^i \in \mathbb{R}^{T \times d_{model}}$ of $i$-th block. This output then becomes the input to the next encoder layer or to the classifier if it is the final layer.

This process repeats at each encoder layer $i = 1$ to $L$. Notably, the same hypernetwork is reused at every layer, receiving different outputs $Y^{i-1}$ as its input $x_i$, and thereby producing different $W^i, b^i$ for each layer. The direct application of $W^i$ to each token provides a more straightforward feed-forward style computation at the $i$-th layer, but since $W^i$ is a function of the entire sequence $Y^{i-1}$, the transformation at layer $i$ is globally context-aware. This weight-sharing across layers reduces the total number of learnable parameters compared to having separate attention parameters per layer while preserving the context-aware ability. To understand how the shared hypernetworks emulate the vanilla multi-head self-attention, we include a centered kernel alignment (CKA) analysis in the Appendix C.

## 5 EXPERIMENTS

In our experiments, we first compare HyperBERT against a BERT (Devlin et al., 2019) baseline of a similar scale. Both HyperBERT and the BERT baseline use a 4-layer Transformer encoder architecture. The baseline is a BERT variant with 4 encoder layers (each with 4 attention heads), whereas HyperBERT replaces each multi-head self-attention mechanism with a dynamically generated MLP. In particular, HyperBERT employs a 4-head, 2-layer Transformer decoder as the shared hypernetwork that generates the weights for single-layer MLPs used in place of the query, key, value, and output projection matrices of attention. To further validate our results, we also compare HyperBERT to several strong MLP-based models from prior work (Mai et al., 2023; Liu et al., 2021; Tolstikhin et al., 2021). Finally, we conduct a transplant experiment to assess the transferability of HyperBERT's learned hypernetwork in a deeper model. Appendix E contains our complete training details and an architectural ablation study.

Table 1: Fine-tuning performance on the test set of GLUE benchmark. Scores are test set accuracies.

| Model | CoLA | SST-2 | MRPC | STS-B | QQP | MNLI (m/mm) | QNLI | RTE | WNLI | AX | Score |
|-------|------|-------|------|-------|-----|-------------|------|-----|------|-----|-------|
| BERT | 8.3 | 85.9 | 79.7/ 70.3 | 71.9/ 68.7 | 66.0/ 85.9 | 58.4/ 59.4 | 84.3 | 55.2 | 65.1 | 10.7 | 64.3 |
| HyperBERT | 9.5 | 85.5 | 78.0/ 69.6 | 73.3/ 71.5 | 65.0/ 85.6 | 59.0/ 61.0 | 81.7 | 57.5 | 62.3 | 14.9 | 64.2 |

## 5.1 PRE-TRAINING AND FINE-TUNING

We pre-trained both HyperBERT and the baseline on the WikiText-103 dataset (Merity et al., 2017). WikiText-103 is a large corpus of over 100 million tokens of verified high-quality Wikipedia articles. We used the standard BERT masked language modeling (MLM) and Next Sentence Prediction (NSP) training objectives for pre-training. We randomly masked 15% of tokens in each input sequence. For a fair comparison, both models were pre-trained for 40 epochs, following the original BERT pre-training procedure.

After pre-training, we fine-tune both models on downstream language understanding tasks using the General Language Understanding Evaluation (GLUE) (Wang et al.) benchmark. We followed standard GLUE fine-tuning practice: for each task, we initialized both models with their pre-trained weights, fine-tuned them on the task's training set for 10 epochs, searched on the validation set for optimal hyperparameters, and finally evaluated the best checkpoints on the test set. The search spaces are shown in Appendix E. We searched for five different random seeds for statistical robustness.

## 5.2 PERFORMANCE

Table 1 summarizes the fine-tuning results on the test set of the GLUE benchmark. Overall, Hyper-BERT achieves an average GLUE score of 64.2, essentially matching the BERT baseline's score of 64.3 while using approximately 6% fewer parameters. We also include the GLUE scores of BERT models at similar scales distilled from the pre-trained original BERT-large model (Turc et al., 2019) in Appendix C.

The results suggest that the hypernetwork-driven feature mixing mechanism can serve as an effective drop-in replacement for multi-head self-attention at this scale, achieving equivalent language understanding capability.

Table 2 reports the results of the comparison with baselines from recent studies. As shown, HyperBERT significantly outperforms baselines directly trained on downstream tasks. Notably, HyperBERT achieves the highest scores on SNLI, QQP, QNLI, and SST-2.

To investigate the performance discrepancy on MNLI, we hypothesized that BERT-style models struggle to learn a rich [CLS] token representation when pre-trained on small-scale datasets such as WikiText-103. To

Table 2: Performance against MLP-based attention alternatives on key benchmarks. Scores are test set accuracies. All shown baselines are directly trained on the downstream task datasets.

| Model | MNLI | SNLI | QQP | QNLI | SST |
|-------|------|------|-----|------|-----|
| MLPMixer | 62.9 | 80.1 | 83.5 | 70.5 | 81.2 |
| gMLP | 61.2 | 80.9 | 82.5 | 60.2 | 79.5 |
| HyperMixer | **66.1** | 81.7 | 84.1 | 77.1 | 81.4 |
| HyperBERT | 60.0 | **84.8** | **85.6** | **81.7** | **85.5** |

test this hypothesis, we trained a 12M-parameter version of HyperBERT directly on MNLI, comparing the classification performance of using the standard [CLS] token versus using a simple mean pooling of all token representations. For comparison, a previous study from Google (Turc et al., 2019) documents the MNLI performance of BERT models with varying sizes that are trained directly on MNLI. The experiment result is shown in Appendix C. HyperBERT with mean pooling achieves 64.0 when directly trained on MNLI. Notably, HyperBERT with mean pooling exceeds the fine-tuning result with fewer parameters, further suggesting that the [CLS] token is suboptimal at this pre-training

Table 3: Transferred 6-layer model fine-tuning performance on the test set of GLUE benchmark

| Model | CoLA | SST-2 | MRPC | STS-B | QQP | MNLI (m/mm) | QNLI | RTE | WNLI | AX | Score |
|---|---|---|---|---|---|---|---|---|---|---|---|
| BERT | 1.2 | 78.9 | 80.5/ 68.6 | 29.0/ 28.2 | 53.7/ 74.7 | 47.0/ 46.4 | 62.3 | 52.5 | 57.5 | 2.0 | 51.8 |
| HyperBERT | 5.4 | 79.8 | 79.9/ 66.5 | 27.0/ 23.2 | 54.8/ 76.7 | 50.6/ 51.8 | 62.3 | 50.1 | 59.6 | 14.4 | 52.5 |

scale. Although our MNLI score is still below HyperMixer's 66.1, we consider our score well within the expected range.

### 5.3 TRANSFERABILITY ANALYSIS

Finally, we investigate the transferability of HyperBERT by transplanting the trained components of each model into a deeper encoder. The goal of this experiment is to test how well the knowledge learned in the original 4-layer models can generalize when the model's depth is increased under a controlled parameter budget. We construct a deeper 6-layer encoder for each model, initialize part of it with the pretrained weights from the 4-layer model, and then fine-tune directly on GLUE to measure the performance. Crucially, we ensure that both models transfer an equal number of parameters from their 4-layer version into the 6-layer model, so that neither has an inherent size advantage.

**HyperBERT transplant.** In HyperBERT's case, we transfer the shared hypernetwork. We took the pre-trained hypernetwork from the 4-layer HyperBERT and injected it into a new 6-layer HyperBERT model. Since HyperBERT's hypernetwork is shared across all layers, we can directly reuse it to generate MLPs for additional layers without adding new hypernetwork parameters. The trained hypernetwork contains 2,567,936 parameters in total, and all of these were transferred. The new 6-layer HyperBERT encoder thus has six MLP-based attention-replacement modules, all driven by the same transferred hypernetwork. The remaining parts of the model were randomly initialized. We then fine-tuned this 6-layer HyperBERT on the GLUE tasks as before.

**Baseline transplant.** For the baseline, we created a 6-layer BERT encoder and initialized it with the pre-trained attention parameters from the 4-layer baseline, but only up to the same 2.567M total parameter count as above. In practice, we copied the complete Q, K, V, O projection matrices from two of the original layers and a portion of the third layer's projections, such that the sum of transferred parameters equals 2,567,936. Notably, the 6-layer HyperBERT has 31.23 M parameters, whereas the corresponding 6-layer BERT baseline has 34.97M. Thus, HyperBERT uses ∼11% fewer parameters at this depth, compared with ∼6% in the 4-layer setting. Scaling HyperBERT to deeper architectures further amplifies HyperBERT's parameter-efficiency advantage.

**Transferability result.** Table 3 reports the performance of the transplanted 6-layer models. The HyperBERT transplant achieves an average GLUE score of 52.5, while the baseline transplant reaches 51.8. Although the overall scores are lower than those of the fully pre-trained 4-layer models, HyperBERT preserves more of its performance advantage when scaled up. These results suggest that the knowledge learned in HyperBERT's hypernetwork is more transferable to a deeper architecture than the knowledge in learned attention weight matrices. In other words, HyperBERT's hypernetwork generalizes more robustly when we increase the model depth than the conventional BERT model. We leave a full exploration of scaling to both shallower and deeper models for future work, but these initial results are promising.

## 6 CONCLUSION

In conclusion, we present HyperBERT, a Transformer encoder that replaces multi-head self-attention with a shared, context-conditioned hypernetwork that generates per-layer MLP weights. Our theoretical analysis and experiments indicate that a parameter-efficient hypernetwork can match the language modeling capability of attention and can surpass it when transplanted to deeper architectures. Future work includes scaling HyperBERT across model sizes, improving computational efficiency, and evaluating the method in other domains.

**Ethics Statement** This paper conducts foundational research aiming to discover the feasibility of replacing the multi-head self-attention with a shared hypernetwork. We foresee no immediate negative societal impact. Our experiments use public datasets under their licenses and do not involve human subjects or private data. All authors have read and will adhere to the ICLR Code of Ethics.

**Reproducibility Statement** We provide the model, training setup, and evaluation protocols in Sections 4 and 5, with hyperparameters, search spaces, and additional ablations in Appendix E. Theoretical assumptions and proofs are provided in Sections 3 and Appendix D. We specify datasets and preprocessing in Section 5. Source code, training scripts, and configuration files will be provided upon acceptance of this paper to enable full reproduction of pretraining, fine-tuning, and transfer experiments.

**Use of Large Language Models (LLMs)** LLMs were used as general-purpose assistants to improve the clarity and grammar of the writing in the manuscript, but not for other purposes.

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

# A  LIMITATIONS

Our method currently applies only to encoder-style Transformers and does not directly extend to decoder-only, autoregressive models. The single-layer generated MLP mixes token positions bidirectionally, which conflicts with the unidirectional causal mask required for left-to-right decoding. Thus, supporting GPT-style generation is nontrivial and requires a substantial redesign of the weight-generation process (e.g., causally conditioned parameter generation or position-aware decoding). Our experiments are also limited in scale. Since multi-head self-attention is externalized into a shared hypernetwork, its architectural and hyperparameter choices can significantly affect parameter efficiency, computational cost, and downstream quality. We plan to scale to deeper encoders and GPT-style decoders, and to systematically explore alternative hypernetwork designs and hyperparameters to identify configurations with the best efficiency–performance trade-offs in our future work.

# B  SOCIETAL IMPACT

HyperBERT demonstrates that a single hypernetwork can provide context-conditioned weights for multiple encoder layers, transforming the multi-head self-attention mechanism into reusable modules rather than fixed, separate units. Such modularity aligns with broader efforts in resource-efficient modeling and can potentially lower the computational and memory barriers that currently restrict cutting-edge NLP to well-funded labs. By achieving Transformer-level accuracy with fewer parameters and transferable weights, HyperBERT shows the potential for richer knowledge transfer across models and configurations. It also suggests a path toward "plug-and-play" language models that smaller organizations can adapt. Moreover, explicit weight generation encourages research on modular architectures that can be updated or extended incrementally rather than replaced, reducing redundant training and associated energy costs.

# C  FURTHER RESULTS

Table 4 shows the results of different BERT and HyperBERT configurations directly trained on MNLI dataset. HyperBERT models in this experiment have 12M parameters. [CLS] uses the first token representation. Mean pooling averages the token states before the classifier. This aligns with our hypothesis that at this pre-training scale, the [CLS] token is a weaker sentence summary than mean pooling. The gap between "HyperBERT + [CLS]" and "HyperBERT + mean pooling" in Table 4 isolates this effect.

Table 5 shows the effective ranks to preserve different levels of cumulative energy. For each pretrained model, we extract *all self-attention projection matrices* per encoder layer (Q, K, V, and O), flatten and concatenate them for that layer, then stack across layers to form $W \in \mathbb{R}^{P \times L}$ (Section 3), where $P = 4d^2$ in the common head-merged view. We compute the singular values $\{\sigma_i\}_{i=1}^{\min(P,L)}$ of $W$ and the cumulative energy curve $E(k) = \sum_{i=1}^{k} \sigma_i^2 / \sum_{i=1}^{\min(P,L)} \sigma_i^2$. $r@X$ is the smallest $k$ such that $E(k) \geq X\%$. We report $r@80$, $r@85$, $r@90$ and $r@90/L$.

Table 6 shows the fitted $\alpha$ and $\beta$ values for equation $r@\text{Threshold} = \alpha \cdot L^\beta$ We fit $r@\text{Threshold} = \alpha L^\beta$ in log–log space using ordinary least squares. Each threshold (80/85/90) is fit independently.

Table 7 shows the GLUE scores of BERT models at different scales. All listed models are distilled from BERT-large, which is pre-trained on a corpus $\sim 50$ times larger than WikiText-103. This table contextualizes our pre-training scale by comparing it to compact BERT variants distilled from BERT-Large (trained on a much larger corpus). These numbers are not directly comparable to our models' absolute scores. Since these distilled models benefit from teacher supervision and vastly larger pretraining data than WikiText-103, their absolute scores can exceed those of models trained from scratch at similar parameter counts. We therefore use them only as sanity checks and not as head-to-head baselines against HyperBERT.

Table 8 shows the diagonal of the resulting CKA similarity matrix. We feed the same set of 6 sentences to both the pre-trained HyperBERT and the BERT baseline. We then compute the CKA similarities between their hidden state representations after each attention block. High similarity at shallow depths (Embed, L1) indicates HyperBERT learns comparable early transformations. The similarity drop at deeper layers (L3–L4) suggests divergent internal computations despite matched end-task accuracy.

Table 4: Comparison of validation accuracies on the MNLI dataset for various BERT models and HyperBERT configurations, all trained directly on the MNLI.

| Model | MNLI |
|---|---|
| BERT-Tiny | 58.8 |
| BERT-Mini | 63.2 |
| BERT-Small | 64.3 |
| BERT-Medium | 66.9 |
| BERT-Base | 67.4 |
| HyperBERT + [CLS] | 58.9 |
| HyperBERT + mean pooling | 64.0 |

Table 5: Empirical validation of the low-rank assumption across various pre-trained Transformers. $L$ denotes the number of layers, and $r@X$ is the effective rank required to preserve $X\%$ of the cumulative energy.

| Model | $L$ | $r@80$ | $r@85$ | $r@90$ | $r@90/L$ |
|---|---|---|---|---|---|
| T5-small | 6 | 4 | 5 | 5 | 0.833 |
| BERT-base-uncased | 12 | 10 | 10 | 11 | 0.917 |
| RoBERTa-base | 12 | 10 | 10 | 11 | 0.917 |
| GPT2-small | 12 | 10 | 10 | 11 | 0.917 |
| facebook/opt-125m | 12 | 10 | 10 | 11 | 0.917 |
| T5-base | 12 | 8 | 9 | 10 | 0.833 |
| google/Gemma-2b | 18 | 14 | 15 | 16 | 0.889 |
| TinyLlama-1.1B-Chat-v1.0 | 22 | 17 | 18 | 20 | 0.909 |
| BERT-large-uncased | 24 | 19 | 20 | 22 | 0.917 |
| RoBERTa-large | 24 | 19 | 20 | 21 | 0.875 |
| GPT2-medium | 24 | 19 | 20 | 22 | 0.917 |
| facebook/opt-350m | 24 | 19 | 20 | 21 | 0.875 |
| T5-large | 24 | 9 | 10 | 13 | 0.542 |
| Qwen1.5-0.5B | 24 | 18 | 20 | 21 | 0.875 |
| google/Gemma-7b | 28 | 22 | 24 | 25 | 0.893 |
| Llama-2-7b-hf | 32 | 25 | 27 | 29 | 0.906 |
| Mistral-7B-v0.1 | 32 | 26 | 27 | 29 | 0.906 |

Table 6: Fitted $\alpha$ and $\beta$ at different cumulative energy threshold

| Threshold | $\alpha$ | $\beta$ |
|---|---|---|
| 80 | 0.769 | 0.990 |
| 85 | 0.916 | 0.954 |
| 90 | 0.907 | 0.985 |

Table 7: GLUE scores of BERT models distilled from BERT-Large.

| Model | CoLA | SST-2 | MRPC | STS-B | QQP | MNLI (m/mm) | QNLI | RTE | WNLI | AX | Score |
|---|---|---|---|---|---|---|---|---|---|---|---|
| BERT-Tiny | 0.0 | 83.2 | 81.1/ 71.1 | 74.3/ 73.6 | 62.2/ 83.4 | 70.2/ 70.3 | 81.5 | 57.2 | 62.3 | 21.0 | 64.2 |
| BERT-Mini | 0.0 | 85.9 | 81.1/ 71.8 | 75.4/ 73.3 | 66.4/ 86.2 | 74.8/ 74.3 | 84.1 | 57.9 | 62.3 | 26.1 | 65.8 |
| BERT-Small | 27.8 | 89.7 | 83.4/ 76.2 | 78.8/ 77.0 | 68.1/ 87.0 | 77.6/ 77.0 | 86.4 | 61.8 | 62.3 | 28.6 | 71.2 |

Table 8: CKA similarity scores comparing the layer-wise hidden states of the pre-trained HyperBERT and BERT models. The diagonal of the similarity matrix is shown. This result suggests that our shared hypernetwork successfully learns the foundational transformations of a standard BERT in its early layers. Although the final fine-tuning scores are similar, HyperBERT and the BERT baseline have significantly different transformations in deeper layers.

| Depth | CKA Similarity |
|---|---|
| Embed vs. Embed | 0.89 |
| L1 vs. L1 | 0.90 |
| L2 vs. L2 | 0.78 |
| L3 vs. L3 | 0.39 |
| L4 vs. L4 | 0.54 |

# D  ADDITIONAL THEORETICAL ANALYSIS

## D.1  SCALABLE PARAMETER EFFICIENCY

Consider the dummy Hypernetwork $H_{dummy}(e_l)$ defined in Section 3.2, where $e_l$ is the layer index embedding. Because $H_{dummy}$ actually stores all the attention weights of the Transformer model, $H_{dummy}$ can approximate the corresponding Transformer model without information loss. The weight matrix $\Theta_{dummy} \in \mathbb{R}^{L \times P}$ of $H_{dummy}(e_l)$ can be low-rank factorized into two matrices $U \in \mathbb{R}^{L \times r}$ and $V \in \mathbb{R}^{r \times P}$ where $r$ is the effective rank. Thus, the dummy hypernetwork can be rewritten as:

$$H_{dummy} = U \cdot V$$

The total number of parameters is now $L \cdot r + r \cdot P$. The efficiency benefit $E(L)$ we get from the low-rank factorization is:

$$E(L) = L \cdot P - (L \cdot r + r \cdot P)$$
$$E(L) = L \cdot P - \alpha \cdot L^{\beta+1} - \alpha \cdot P \cdot L^{\beta}$$
$$E(L) = LP(1 - \frac{\alpha L^{\beta}}{P} - \alpha L^{\beta-1})$$

Since $0 < \alpha < 1, 0 < \beta < 1$, the term $L^{\beta-1} < 1$, and the term $\frac{\alpha L^{\beta}}{P}$ is negligible as the number of parameters per layer $P$ is much larger than $L^{\beta}$. Thus, the efficiency benefit remains positive, ensuring that $E(L) > 0$ for all practical model depths.

## D.2  THE CAPACITY OF THE ONE-LAYER MLP

Theoretically, our design is grounded in the reformulation of Multi-head Attention (MHA) itself as an implicit linear hypernetwork. As shown in previous studies (Schug et al., 2025), the final transformation in MHA is a linear combination of the value vectors. For a given input sequence $X$, the operation can be expressed as:

$$MHA_q(X) = \sum_{k=1}^{T} (\sum_{h=1}^{H} a_{h,q,k} W_h^{out} W_h^{value}) x_k$$

In this expression, $a_{h,q,k}$ is the attention score of head $h$, and $X = [x_1, \ldots, x_T]$, where $T$ is the sequence length:

$$a_{h,q,k} = \sigma(\frac{Q_h^\top K_h}{\sqrt{D^{head}}})$$

In here, the attention score of head $h$ plays the role of the input to the Hypernetwork $W_h^{out} W_h^{value}$. The final step of $MHA(X)$ can be written as:

$$MHA_q(X) = \sum_{k=1}^{T} \underbrace{W_{q,k}}_{\text{value network}} x_k$$

Where $W_{q,k}$ is a matrix dynamically computed from the non-linear query-key interactions. This demonstrates that the core MHA operation applied to the input sequence $X$ is a dynamically-weighted linear transformation.

Our HyperBERT architecture externalizes this process. The complexity is moved from the implicit attention score calculation to our explicit, non-linear hypernetwork, which generates the MLP weights based on the full sequence context. The application of these weights to the input tokens remains a single linear transformation followed by a non-linear activation. Therefore, a one-layer MLP is theoretically sufficient to emulate the functional form of the MHA mechanism.

### D.3 THEORETICAL COMPUTATIONAL COMPLEXITY

**Symbols.** $T$: sequence length, $d$: model width, $d_{\text{ff}} = 4d$: feed-forward width, $h$: hypernetwork width.

**Floating point operations (FLOPs) per block of Baseline (BERT).**

$$\text{FLOPs}_{\text{bert}} = \underbrace{3Td^2}_{\text{Q,K,V proj}} + \underbrace{2T^2d}_{\text{attn scores \& mix}} + \underbrace{Td^2}_{\text{out proj}} + \underbrace{2Tdd_{\text{ff}}}_{\text{FFN}} = \boxed{12\,Td^2 + 2\,T^2d}.$$

**Our HyperBERT per layer.** "Attention" replacement matmul $xW$: $Td^2$; FFN: $2Tdd_{\text{ff}} = 8Td^2$. Shared hypernetwork (2-layer, 4-head decoder; generates $d$ rows twice: weight & bias) with full-sequence memory yields

$$\text{FLOPs}_{\text{hyper}} = \underbrace{T(2dh + 2h^2)}_{\text{context MLP}} + \underbrace{2\big(2d^2h + 2dTh\big)}_{\text{2-layer decoder, twice}} + \underbrace{2(d^2h + dh^2)}_{\text{row projections}},$$

Hence, the hyperBERT layer cost is

$$\boxed{\text{FLOPs}_{\text{Hyper}} = 9\,Td^2 + 6d^2h + 6dTh + 2dh^2 + 2Th^2}$$

**Crossover (when HyperBERT beats BERT in asymptotics).** To make HyperBERT more computationally efficient than the BERT baseline, we need to solve the following inequality:

$$\text{FLOPs}_{\text{bert}} - \text{FLOPs}_{\text{hyper}} > 0 \tag{10}$$

Since T must be a positive integer, we have the following solution:

$$T > \frac{-3d^2 + \sqrt{9d^4 + 12d^3h + 40d^2h^2 + 24dh^3 + 4h^4} + 6dh + 2h^2}{4d} \tag{11}$$

Figure 2 plots the per-layer FLOPs for the BERT baseline and our HyperBERT layer under the counting conventions used throughout this section. These FLOPs conventions follow the standard Transformer.

## E ADDITIONAL EXPERIMENT DETAILS

### E.1 EXPLORATION

We have explored different model architectures, and the current implementation of HyperBERT performs the best. Below is a list of different architectures we have explored.

1. **Variant 1** For variant 1, we use separate hypernetworks for different encoder layers. The input context is the attention score from $\frac{QK}{\sqrt{d_dk}}$. The input is further compressed into a $d_{model}$ vector. The hypernetworks generate weights for the Value and Output projection matrices only. Also, instead of generating weights chunk-wise, the hypernetworks generate all weights of the MLP at once.

2. **Variant 2** Variant 2 changed the following setups compared to variant 1. $(i)$ The hypernetwork takes the input of attention sub-layers as input instead of attention scores. $(ii)$ The hypernetwork is learned to generate the weights for all four Q, K, V, O projection matrices with different layer-type embedding and projection layers.

3. **Variant 3** Variant 3 uses one shared MLP across layers to generate weights and biases for one MLP to replace the multi-head self-attention sub-layer. The weights and biases are generated at once instead of chunk-wise. All Variant 1,2,3 are larger than BERT baseline.

4. **Variant 4** Variant 4 uses one shared MLP across layers to generate weights and biases for one MLP to replace the multi-head self-attention sub-layer. Variant 4 generates weights and biases chunk-wise. This approach significantly reduces the size of the output space and makes the size of variant 4 smaller than the BERT baseline. However, the input of the hypernetwork is still compressed into a $d_{model}$-dimension vector.

5. **Variant 5** Variant 5 is the HyperBERT implementation presented in this paper. Compared to previous variants, variant 5 uses the full input matrix as its context without compression.

6. **Variant 6** Variant 6 removes the context encoder compared to variant 5.

Table 9: Final pre-training losses of all variants and baseline

| V1 | V2 | V3 | V4 | V5 | V6 | BERT |
|------|------|------|------|------|------|------|
| 3.85 | 7.91 | 5.78 | 4.80 | 2.81 | 3.36 | 3.06 |

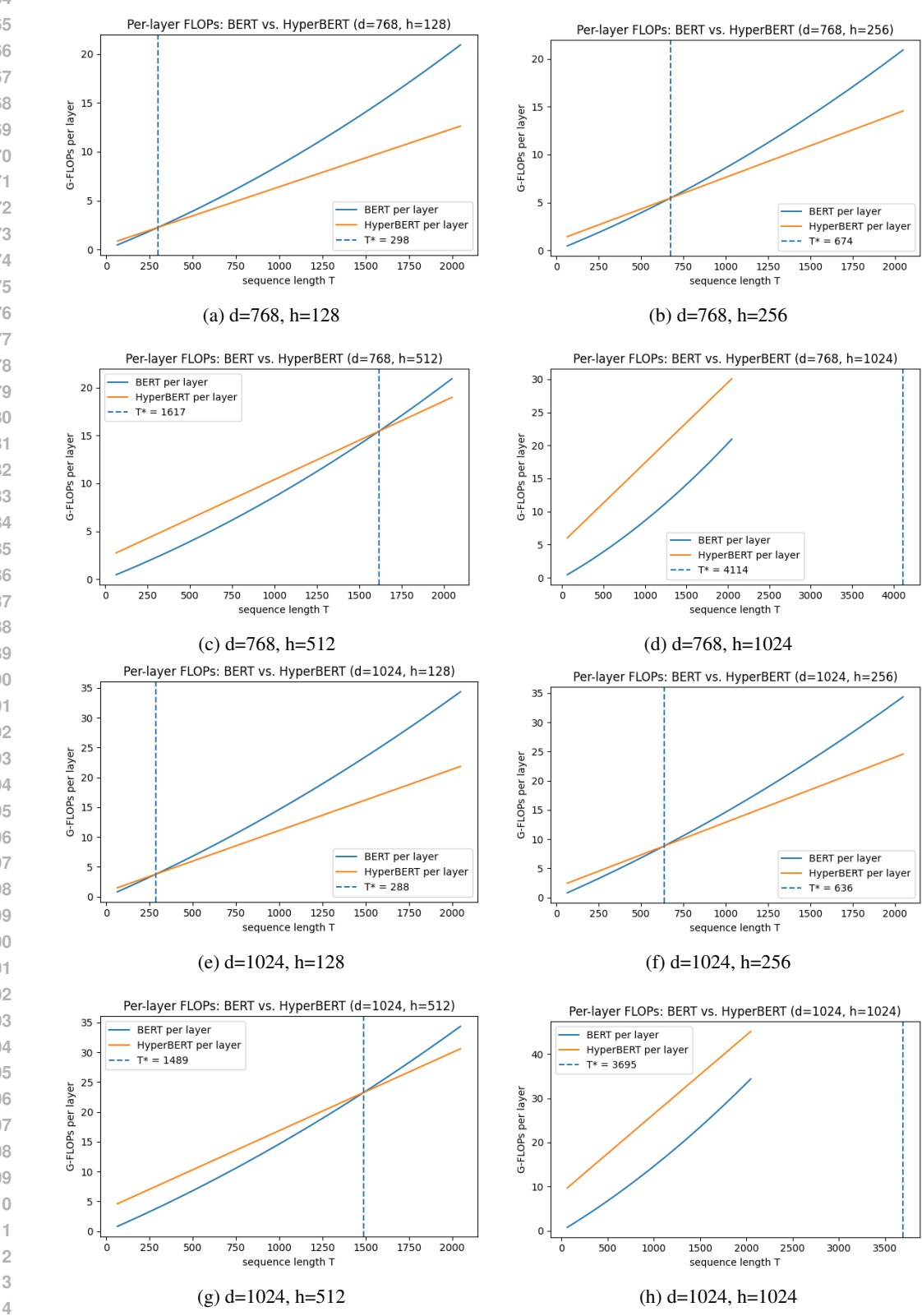

Figure 2: Per-layer theoretical FLOPs (BERT vs. HyperBERT) across configurations. Each panel plots the sequence length $T$ vs. FLOPSs. Vertical dashed lines mark the crossover $T^\star$ for each setting.

We fully pre-trained five additional HyperBERT variants. Due to resource limits, we could not complete a full grid search over GLUE for all variants. To determine the optimal architecture, we evaluated all HyperBERT variants on two criteria: final pre-training loss and preliminary fine-tuning performance on key GLUE tasks. As shown in Table 9, Variant 5 achieved the lowest pre-training loss after 40 epochs. Given its superior performance in both pre-training and the initial fine-tuning tests, we selected Variant 5 as the final HyperBERT implementation for all experiments in this paper.

### E.2 PRE-TRAINING

During pretraining we adopt the original BERT MLM and NSP objectives. HyperBERT and the BERT baseline are each trained for 40 epochs with a learning rate of $1 \times 10^{-4}$. Training uses the AdamW optimizer ($\beta_1 = 0.9$, $\beta_2 = 0.999$) with an $L_2$ weight decay of 0.01, GELU activations, and a dropout rate of 0.1 on every layer. HyperBERT is trained with a batch size of 20, while the BERT baseline uses 24; all remaining hyperparameters are identical. Unlike the original BERT setup, every model has a maximum sequence length of 256 and an embedding dimension of 512, and the sequence length is kept at 256 throughout pretraining. HyperBERT uses a context dimension $d_c = 256$. All runs include 10 000 warmup steps. All pre-trained variants and baselines converge after 40 epochs.

### E.3 FINE-TUNING

For fine-tuning, most tasks share the same search space on the validation set.

- **Learning rate**: 1e-6, 2e-6, 3e-6, 5e-6, 7e-6, 1e-5, 3e-5, 5e-5, 7e-5, 1e-4,
- **Random seed**: 39, 40, 41, 42, 43
- **Batch size**: 8 16 20

For CoLA, the loss function has class weights of $[2.5, 1]$ for the imbalanced training set. The batch size is set to 64.

For fair comparison, the search spaces for the same task are identical for all pre-trained models. Table 10 shows the mean $\pm$ standard deviation over five different seeds. For tasks with multiple metrics, we report the average score of all metrics.

Table 10: Mean and standard deviation of the best validation scores achieved across runs with different random seeds.

| Model | CoLA | SST-2 | MRPC | STS-B | QQP | MNLI (m/mm) | QNLI | RTE | WNLI |
|---|---|---|---|---|---|---|---|---|---|
| BERT | 10.09 $\pm$ 1.21 | 87.18 $\pm$ 0.50 | 78.74 $\pm$ 0.52 | 79.31 $\pm$ 0.80 | 86.17 $\pm$ 0.59 | 55.47 $\pm$ 4.28 | 84.06 $\pm$ 0.33 | 61.59 $\pm$ 2.55 | 53.24 $\pm$ 9.09 |
| HyperBERT | 11.96 $\pm$ 4.61 | 87.00 $\pm$ 0.43 | 79.66 $\pm$ 0.58 | 80.37 $\pm$ 0.52 | 85.83 $\pm$ 0.28 | 59.00 $\pm$ 0.65 | 81.74 $\pm$ 0.49 | 60.94 $\pm$ 0.74 | 54.93 $\pm$ 3.98 |

### E.4 TRANSFERABILITY EXPERIMENT

We first calculate the total number of parameters in the pretrained hypernetwork and use these weights to initialize a six layer HyperBERT. Next, we copy the same number of attention weights from the four layer BERT baseline to construct a six layer BERT baseline. Because HyperBERT learns attention across all layers within a single shared hypernetwork, its parameter count grows more slowly than that of the BERT baselines. In the four-layer setting, HyperBERT has approximately 6% fewer parameters than the BERT baseline, and in the six-layer setting, it has about 11% fewer. Most tasks share the same search space on the development set.

- **Learning rate**: 1e-6, 3e-6, 5e-6, 7e-6, 1e-5, 3e-5, 5e-5
- **Random seed**: 42
- **batch size**: 20

For QNLI and CoLA, we set the batch size to 16. For CoLA, the loss function also has class weights of $[2.5, 1]$.

### E.5 COMPUTATIONAL RESOURCE

All pre-training can be finished in under 80 hours on a Linux workstation with 128 GiB of RAM and two NVIDIA RTX 4090 GPUs. On this workstation, fine-tuning and transferability experiments for every model conclude within 48 hours across all tasks. For faster exploratory runs, we also used a Linux server equipped with 1800 GiB of RAM and eight NVIDIA A100 GPUs.

