# OpenReview forum: "Learn All You Need in One Hypernetwork"
_ICLR.cc/2026/Conference — Submitted to ICLR 2026_

### Official Review · Reviewer_NJeA · 2025-10-27

**Soundness:** 1
**Presentation:** 1
**Contribution:** 1
**Rating:** 0
**Confidence:** 5

**Summary:**

This work proposes HyperBERT, which replaces the self-attention mechanism in the encoder-only transformer (BERT) with an MLP whose weights are generated by a hypernetwork parameterized as a decoder-only transformer. Experiments compare 4-layer BERT and HyperBERT models, using WikiText-103 (100M tokens) for pre-training and the GLUE benchmark for evaluation.

**Strengths:**

The reviewer found no notable strengths in this submission.

**Weaknesses:**

There are three major issues: (1) the soundness and relevance of the proposed method are weak; (2) the practical significance of the results is very limited; and (3) the presentation lacks mathematical rigor.

Method: While BERT played an important role in the early stage of pre-trained language modeling (~2018–2020), it has not remained the mainstream paradigm, primarily because it is not a standalone model that can be used as-is---unlike the more versatile GPT-style decoder-only language models. Given this context, there should be a strong and specific motivation for revisiting the BERT setting; however, the reviewer did not find such a compelling justification in this work.

The main idea of using an MLP with context-dependent weights for sequence processing is also not new---as acknowledged by the authors through their citations of Schlag et al. (2021) and Schug et al. (2025).

Experiment: The experimental setting is extremely limited. One of the main claims in the abstract is that *"prior work has been limited to small-scale tasks"*; however, pre-training on the 100M-token WikiText-103 dataset also clearly falls within the small-scale regime by today’s standards. Even in academic settings, it is now common to train models with up to 1B parameters on roughly 10B tokens to meaningfully evaluate modern language modeling methods. Additionally, as noted above, the baseline comparison is restricted to BERT. There is no clear justification for this choice, given that more general decoder-only language models can perform similar tasks.

Presentation/Clarity: There are also issues with mathematical rigor, notation, and overall clarity. For example, in Sec. 3.1, two different notations—$f_l$ and $F$—are introduced in Equation (2) for the same function, and their usage is inconsistent (Sec. 3.1 uses $F$, while Sec. 3.2 uses $f_l$). In Sec. 3.2, the phrase *"that is one-hot at position $l$"* likely refers to the $l$-th coordinate, not the position. In Sec. 3.4, *"Basis memory $U$ that is shared"* appears without a clear transition; some connection to Sec 3.3. needs to be clarified more smoothly. The statement *"Its output dimension is P, so it can replace $w_l$ inside F"* uses non-italicized mathematical symbols ($P$, $F$), which is inconsistent with the rest. Similarly, *"the input is a layer code $c_l$"*---$c_l$ should be mathematically defined, including its dimensionality if it is a vector.

More broadly, given that this theoretical result is not particularly strong---hypernetworks that are properly parameterized by neural networks are naturally universal approximators themselves (and the low-rank observations motivate parameter-sharing)---the details would be more appropriate for the appendix.

Overall, this submission falls well below the standards expected at general machine learning conferences.

**Questions:**

The reviewer has no further questions and considers it unlikely that this work will become acceptable after any rebuttal or discussion.

One relevant missing reference:
HyperTransformer: Model Generation for Supervised and Semi-Supervised Few-Shot Learning
Andrey Zhmoginov, Mark Sandler, Max Vladymyrov. ICML 2023.

---

> ### Author Response · Authors · 2025-11-15
>
> We thank Reviewer NJeA for their time. However, we must respectfully clarify several key points where the review's summary appears to diverge from the verifiable contents of our paper and prior works. These discrepancies, which concern our core novelty, theoretical contributions, and mathematical definitions, form the basis of the 0-point rating. We clarify them below.
>
> ### **1. Mischaracterization of Our Novelty and Prior Work**
>
> Reviewer NJeA states that "The main idea of using an MLP with context-dependent weights for sequence processing is also not new---as acknowledged by the authors through their citations of Schlag et al. (2021) and Schug et al. (2025)."
>
> We have never claimed to invent the idea of "using an MLP with context-dependent weights." Our novel contribution, which Reviewer NJeA does not address, is the specific architecture of a BERT-style Transformer encoder variant in which multi-head self-attention is entirely replaced by MLPs generated by an explicit, shared hypernetwork. We pre-train this model and fine-tune on GLUE. This is stated clearly throughout our paper.
>
> Moreover, **the cited works do not claim their contribution is using an MLP with context-dependent weights for sequence processing.** What the cited works actually cover are:
>
> 1. Schlag et al. (2021) claim their contribution is **showing the "formal equivalence of linearised self-attention mechanisms and fast weight controllers (FWPs)".** They do **not** propose replacing the full Q/K/V/O matrices in a BERT model with an explicit, shared hypernetwork.
>
> 2. Schug et al. (2025) claim their contribution is to **"reformulate standard multi-head attention from a hypernetwork perspective"** and **"introduce a simple modification to multi-head linear attention".** The proposed HYLA in their paper is intended to verify their hypothesis, which the authors state: "Specifically, we define Hypernetwork Linear Attention (HYLA) to use a single hidden layer value network by adding a nonlinearity. This should increase the expressivity of the subfunctions that are learned and recomposed by the layer." HYLA does not train an explicit or shared hypernetwork that generates weights for the model. Their amplified hypernetwork mechanism remains layer-local.
>
> Our work **externalizes** the multi-head self-attention mechanism into a concrete, trainable, and shared module, and is, to our knowledge, the first to demonstrate its viability in a pre-training and fine-tuning setting.
>
> ### **2. Inaccurate Summary of Our Theoretical Contribution**
>
> Reviewer NJeA claims "More broadly, given that this theoretical result is not particularly strong---hypernetworks that are properly parameterized by neural networks are naturally universal approximators themselves (and the low-rank observations motivate parameter-sharing)---the details would be more appropriate for the appendix."
>
> This is **not** our theoretical result. Our theoretical analysis in Section 3 is not a generic universal approximation theorem. We specifically prove the existence of a shared, parameter-efficient hypernetwork that can approximate the set of layer-wise attention transformations within a bounded error under the low-rank assumption.
>
> ### **3. Misunderstanding of Our Motivation and Scope**
>
> Reviewer NJeA claims "While BERT played an important role in the early stage of pre-trained language modeling (~2018–2020), it has not remained the mainstream paradigm, primarily because it is not a standalone model that can be used as-is---unlike the more versatile GPT-style decoder-only language models. Given this context, there should be a strong and specific motivation for revisiting the BERT setting; however, the reviewer did not find such a compelling justification in this work."
>
> Our work focuses on encoder-only models, which remain critical for representation learning, information retrieval, and other areas. We are explicit in our Limitations section that the current HyperBERT design does not directly transfer to decoder-only Transformers because our shared hypernetwork, in its present form, conditions on information that depends on the entire input sequence. Focusing on encoder-only models is a legitimate research choice.
>
> Regarding scale, we do **not** claim to beat SOTA LLMs in our paper. Instead, we present feasibility and scaling signals. When scaling up HyperBERT, the key consideration is expressivity. A fixed-capacity hypernetwork cannot support an infinitely deep model. However, our generator-executor paradigm makes this trade-off explicit and tunable, which we believe is a promising direction for scaling future architectures. Our experimental scale is in line with or larger than scale of papers we cite as baselines and related works.

---

> > ### Author Response · Authors · 2025-11-15
> >
> > ### **4. Incorrect Claims About Mathematical Rigor**
> >
> > Reviewer NJeA claims we "introduce two notations in Equation (2) for the same function," misuse "one-hot at position," and present "layer code $c_l$" without mathematical definition.
> >
> > 1. Our Equation (2) defines the layer-$l$'s forward map $f_l$ as the instantiation of the parametric attention map $F$ at weights $w_l$: $$f_l : \mathbb{R}^{T \times d_{model}} \;\longrightarrow\; \mathbb{R}^{T \times d_{model}},\qquad x \;\mapsto\; F(x;\,w_l).$$ Thus, $f_l:=F(\cdot;w_l)$. **They are two different functions:** a family $F(\cdot;w)$ and its layer-specific instance $f_l$. In Section 3.4, we then introduce the shared hypernetwork $H_\theta(c_l)=UG_\phi(c_l)$ whose output dimension matches $P$ "so it can replace $w_l$ inside $F$", reinforcing that $F$ is the single attention map into which either $w_l$ or $H_\theta(c_l)=UG_\phi(c_l)$ is plugged.
> >
> > 2. In Section 3.1, we introduce $l$ as the **layer index** ("for every layer $l \in \{1,\dots,L\}$ ..."), and in Section 3.2 we define $e_l \in \mathbb{R}^L$ as the **one-hot embedding at position $l$**. Thus $l$ is unambiguously the layer index throughout, and $e_l$ its corresponding one-hot embedding.
> >
> > 3. Immediately after introducing "the input is a layer code $c_l$," we state that $c_l$ **can be an embedding of layer index $l$** (the same one-hot embedding $e_l$ from Section 3.2) **or a learned summary of the current hidden state,** giving flexibility in how the hypernetwork is conditioned. This is explicit in our text where we define the basis-coefficient construction and the map $H_\theta(c_l)=UG_{\phi}(c_l).$
> >
> > We appreciate the typography note (e.g., italicizing $P$) and will polish the notation and add a brief bridging sentence in Section 3.4 as a transition.
> >
> > ### **5. Missing Reference**
> >
> > We thank Reviewer NJeA for suggesting HyperTransformer (*Zhmoginov et al., 2023*). We will add HyperTransformer to our Related Work section as suggested. However, this paper uses a Transformer as a hypernetwork to generate CNN weights for few-shot vision tasks. It is not an investigation into the hypernetwork mechanism within language models. Therefore, it is not directly related to our work.
> >
> > We hope these clarifications are helpful.

---

> > > ### Comment · Reviewer_NJeA · 2025-11-17
> > >
> > > I sincerely thank the authors for their response.
> > >
> > > I would like to emphasize that I rated this work with a score of 0 and confidence 5---which reflects a strong assessment that this submission is far below the acceptance threshold for ICLR. This was not a light decision but an honest and considered evaluation.
> > > As a reviewer, it is my responsibility to provide accurate feedback and clearly indicate when a submission does not meet the expected standards. In my view, this is such a case, given the presented method and experimental results.
> > >
> > > The hypernetwork is a well-established concept---parameterizing or generating the weights of a neural network via another neural network---and it has been widely applied across architectures.
> > > Applying the hypernetwork concept to a specific model (e.g., MLP layers in BERT) alone does not constitute a strong novelty unless it brings a substantially new insight or produces compelling experimental evidence, which is not the case here.
> > >
> > > Demonstrating that such a method is feasible is not, by itself, a significant contribution.
> > > By no means, I'm asking to report state-of-the-art results. However, as I explained in my original review, the scale of the experiments presented here is far too small given the current academic standards.
> > >
> > > Relatedly, regarding the HyperTransformer reference, the authors state:
> > >
> > > > *However, this paper uses a Transformer as a hypernetwork to generate CNN weights for few-shot vision tasks. It is not an investigation into the hypernetwork mechanism within language models. Therefore, it is not directly related to our work.*
> > >
> > > I would argue that this interpretation is too narrow for a general machine learning venue like ICLR. While *ACL or EMNLP may have language/domain-specific scopes, ICLR’s focus is broader. From that perspective, HyperTransformer---which also uses a Transformer as a hypernetwork to generate weights for another network (CNN)---is clearly a directly related work.
> > >
> > > Finally, while I understand that the authors may strongly defend their position, using phrases such as "Mischaracterization", "Inaccurate Summary", or "Misunderstanding" is less productive in this context. Such claims would be more meaningful in the presence of substantial novelty or nontrivial conceptual complexity, which I do not find here. It is also regrettable that the authors reduce the reviewer’s effort in pointing out confusing notations to a simple "typography note" and even "Incorrect Claims" rather than recognizing it as feedback to improve the manuscript---such acknowledgment is an essential part of professional scientific exchange.
> > >
> > > Given all these points, I will maintain my original rating.

---

> > > > ### Author Response · Authors · 2025-11-18
> > > >
> > > > We appreciate Reviewer NJeA's further engagement. Since the reviewer has clearly stated that they will maintain a score of 0, our goal in this comment is not to seek a change of score. However, we would like to take this opportunity to provide further clarifications for future readers.
> > > >
> > > > ### **1. Clarification on Contribution**
> > > >
> > > > In the original review, our work was summarized as:
> > > >
> > > > >  "The main idea of using an MLP with context-dependent weights for sequence processing $\dots$"
> > > >
> > > > In the latest comment, it is summarized as:
> > > >
> > > > > "Applying the hypernetwork concept to a specific model (e.g., MLP layers in BERT) alone $\dots$"
> > > >
> > > > Both descriptions differ from what we actually do and claim.
> > > >
> > > >
> > > > 1. We do **not** claim to invent the idea of "using an MLP with context-dependent weights for sequence processing."
> > > >
> > > > 2. We do **not** apply hypernetworks to MLP/FFN sublayers in BERT in our paper. Our focus is an **explicit, shared hypernetwork** that replaces the **multi-head self-attention mechanism (Q/K/V/O projections)**, widely used in Transformer architectures, across all layers of a BERT-style encoder.
> > > >
> > > > 3. We provide a constructive theorem showing that a shared hypernetwork can approximate the stack of layer-wise multi-head self-attention transformations with fewer parameters. We study this design empirically in pre-training and fine-tuning settings.
> > > >
> > > > ### **2. Soundness and Notation**
> > > >
> > > > The initial review justified "poor soundness" and "poor presentation" by arguing the following: (*i*) our theoretical result is a generic universal approximation statement; (*ii*) we introduce "two notations for the same function;" and (*iii*) we leave the input layer code $c_l$ mathematically undefined.
> > > >
> > > > We clarified them in our earlier responses. In the latest comment, Reviewer NJeA does not contest these clarifications. The focus is mainly on novelty and experimental scale.
> > > >
> > > > We hope these are helpful for future readers.

---

### Official Review · Reviewer_mtKV · 2025-10-28

**Soundness:** 3
**Presentation:** 3
**Contribution:** 2
**Rating:** 4
**Confidence:** 5

**Summary:**

The papers proposes to use a single shared hypernetwork for all attention layers. It demonstrates that HyperBERT saves parameters and improves performance in downstream tasks.

**Strengths:**

- Well motivated, interesting idea
- Quantifiable, although minor benefits
- explicit discussion of limitations, clearly written

**Weaknesses:**

- Considering that attention as a hypernetwork has already been implemented and the contribution is limited to having a shared hypernetwork between layers, it makes it somewhat incremental
- The model is limited to encoder transformers
- FLOP curves scale worse than standard BERT in terms of sequence length, furthermore, wall-clock time has not been reported which raises a question how fast the networks are in practical implementation
- The scale of the experiments is too small to conclude about the behaviour in larger models
- Parameters saving over BERT are very marginal especially that the comparison is against basic BERT. AlBERT[1] achieves strong results while sharing a lot of layers. HyperBERT effectively does this as well, however, it would not beat the AlBERT.

[1] - **ALBERT: A Lite BERT for Self-supervised Learning of Language Representations**

**Questions:**

Could you include the wall-clock time comparisons to other models, as well as sizes of all models compared?

---

> ### Author Response · Authors · 2025-11-15
>
> We thank Reviewer mtKV for their time and feedback. We are pleased to address the concerns raised.
>
> ### **1. Contribution and Novelty**
>
> Attention as a hypernetwork (*Schug et al., 2025*) does not implement any form of explicit hypernetwork. The authors claim their contribution is to "reformulate standard multi-head attention from a hypernetwork perspective" and "introduce a simple modification to multi-head linear attention". The proposed HYLA in their paper is intended to verify their hypothesis, which the authors state as follows: "Specifically, we define Hypernetwork Linear Attention (HYLA) to use a single hidden layer value network by adding a nonlinearity. This should increase the expressivity of the subfunctions that are learned and recomposed by the layer." HYLA does not train an explicit or shared hypernetwork that generates weights for the model. Their amplified hypernetwork mechanism remains layer-local.
>
> By contrast, our work explicitly externalizes the attention mechanism into one shared generator that dynamically generates per-layer MLP operators conditioned on the current input sequence. This goes beyond prior implicit or layer-local hypernetwork views.
>
> Theoretically, we show that a compact shared hypernetwork can approximate the stack of per-layer attentions with fewer parameters, and explain why increasing depth and context length amplifies both parameter and runtime/memory savings.
>
>
> ### **2. Transferring to Decoder-Only Models**
>
> As noted in our paper's limitations, the current HyperBERT design does not directly transfer to decoder-only transformers because our shared hypernetwork, in its present form, conditions on information that depends on the entire input sequence. When the same generator is invoked multiple times across layers, layer calls could access future tokens and violate causality even with a standard causal mask. With a shared hypernetwork, the conditioning itself must be causal.
>
> However, this implicit hypernetwork mechanism is inherent to both encoder-only and decoder-only architectures. As ongoing work, we are pursuing two causal designs: (*i*) A shared hypernetwork that can provide causal MLPs for token mixing. Each token embedding is updated with information up to that token. (*ii*) A design based on Mamba-style backbone model (*Gu & Dao, 2024*). Mamba also includes this implicit hypernetwork mechanism to improve traditional SSM models (by making the $B_t, C_t, \Delta_t$ matrices dependent on the input token $x_t$) and achieves state-of-the-art performance. This architecture aligns with our generator-executor paradigm and keeps causality. We believe the hypernetwork mechanism is critical for language modeling. We are actively exploring efficient methods for transferring HyperBERT to modern generative language models. The experience with HyperBERT is valuable.

---

> ### Author Response · Authors · 2025-11-15
>
> ### **3. FLOPs, Wall-Clock Time, and Memory Usage**
>
> As shown in our theoretical analysis and Figure 2, HyperBERT's FLOPs curves scale **better** than standard BERT in terms of sequence length. While HyperBERT has a higher fixed cost at short sequences, that fixed cost is amortized and HyperBERT becomes faster as sequence length grows. We will improve the figure to make this more explicit and avoid confusion.
>
> We benchmarked the BERT baseline and HyperBERT models under the same setting on an RTX 4090 GPU, running 1000 iterations per configuration with batch size equal to 1. We report the average runtime per sample and the peak GPU memory usage. Full results are in the tables below.
>
> 4-layer models, $d_{model}= 768$, $h=128$:
>
> | Model | Mode | Sequence length | Avg ms/sample | VRAM Usage (GB)|
> |-|-|-|-|-|
> |BERT|Train|2048|20.96|2.31|
> |HyperBERT|Train|2048|27.92|1.72|
> |BERT|Train|3072|33.68|3.42|
> |HyperBERT|Train|3072|28.49|2.26|
> |BERT|Train|4096|49.22|4.76|
> |HyperBERT|Train|4096|29.46|2.79|
> |BERT|Train|6144|106.04|8.91|
> |HyperBERT|Train|6144|40.37|4.42|
>
> | Model | Mode | Sequence length | Avg ms/sample | VRAM Usage (GB)|
> |-|-|-|-|-|
> |BERT|Inference|2048|4.76|0.88|
> |HyperBERT|Inference|2048|7.88|0.82|
> |BERT|Inference|3072|8.59|1.19|
> |HyperBERT|Inference|3072|8.45|1.13|
> |BERT|Inference|4096|13.07|1.49|
> |HyperBERT|Inference|4096|9.08|1.43|
> |BERT|Inference|6144|29.30|2.67|
> |HyperBERT|Inference|6144|11.17|2.17|
>
>
> 8-layer models, $d_{model}=1024$, $h=128$:
>
> | Model | Mode | Sequence length | Avg ms/sample | VRAM Usage (GB)|
> |-|-|-|-|-|
> |BERT|Train|2048|43.50|4.24|
> |HyperBERT|Train|2048|54.21|2.87|
> |BERT|Train|3072|69.84|6.15|
> |HyperBERT|Train|3072|55.63|3.59|
> |BERT|Train|4096|101.59|8.48|
> |HyperBERT|Train|4096|56.01|4.30|
> |BERT|Train|6144|194.81|14.62|
> |HyperBERT|Train|6144|62.27|5.75|
>
> | Model | Mode | Sequence length | Avg ms/sample | VRAM Usage (GB)|
> |-|-|-|-|-|
> |BERT|Inference|2048|9.18|1.32|
> |HyperBERT|Inference|2048|14.34|1.14|
> |BERT|Inference|3072|17.14|1.63|
> |HyperBERT|Inference|3072|15.15|1.44|
> |BERT|Inference|4096|26.83|1.96|
> |HyperBERT|Inference|4096|16.02|1.75|
> |BERT|Inference|6144|54.46|2.96|
> |HyperBERT|Inference|6144|17.21|2.36|
>
> As described in our paper, introducing the shared hypernetwork adds a fixed hypernetwork cost. Thus, at short sequences, HyperBERT is slower than vanilla BERT. As sequence length grows, that fixed cost is amortized and HyperBERT becomes faster. On memory, HyperBERT consistently reduces peak VRAM usage during training and inference, especially in the deeper models.
>
> ### **4. Scaling to Larger Models**
>
> Our method's advantages in both parameter and compute efficiency amplify as models grow in depth and sequence length.
>
> The first scaling axis is model depth. Our transferability experiment was designed to test this. When we transplanted the same shared hypernetwork from the 4-layer model into a deeper 6-layer model, the parameter savings increased from 6% to 11% . This demonstrates that decoupling the parameter generation from the network's depth leads to efficiency gains as layers are added. This shows a clear path to more significant gains in larger-scale models.
>
> The second scaling axis is sequence length. Our theoretical FLOPs analysis in Appendix shows that HyperBERT's computational cost scales better with sequence length than standard BERT.
>
> The tables we show above further validate that the advantages of HyperBERT amplify as models grow in depth and sequence length.
>
> When scaling up HyperBERT, the key consideration is expressivity. A fixed-capacity hypernetwork cannot support an infinitely deep model. Scaling up HyperBERT requires determining the optimal proportion of total parameters to allocate to the hypernetwork. However, our generator-executor paradigm makes this trade-off explicit and tunable, which we believe is a promising direction for scaling future architectures.

---

> ### Author Response · Authors · 2025-11-15
>
> ### **5. Comparison to ALBERT**
>
> ALBERT (*Lan et al., 2019*) and HyperBERT address orthogonal efficiency axes: ALBERT reduces parameters via static cross-layer weight sharing and embedding factorization. The per-layer operator is fixed. HyperBERT removes layer-local attention weights and uses a shared generator to produce input-conditioned operators across layers. HyperBERT could be combined with ALBERT-style factorization and shared FFNs.
>
> While ALBERT achieves better performance than BERT with fewer parameters, this efficiency comes at a high computational cost as shown in their Table 2. Because its parameters are reused across multiple blocks, ALBERT's total FLOPs are higher than BERT's. This explains why ALBERT-xxlarge, despite having fewer parameters, is reportedly three times slower than the larger BERT baseline.
>
> Moreover, theoretically, there is nothing stopping the hypernetwork from generating the weights for FFNs as well. We are actively exploring this, and our preliminary results are promising. We find that a single, shared hypernetwork can generate parameters to replace both the multi-head self-attention and the subsequent FFNs, unifying the entire Transformer block into a dynamically generated operator.
>
> This design enhances modularity and unlocks more flexible architectures such as adaptive depth. The shared hypernetwork could learn to dynamically generate computational blocks based on context and decide whether to continue processing or to exit early.
>
> We will add further explanation regarding ALBERT in our camera-ready version.
>
> ### **6. Wall clock time and Sizes of all models compared**
>
> Wall-clock time comparisons under identical settings are given above.
>
> Regarding model size, the 4-layer BERT baseline has 28.66M parameters and 4-layer HyperBERT has 27.03M parameters. The 6-layer BERT baseline has 34.97M parameters and 6-layer HyperBERT has 31.23M parameters. We also trained a 12M HyperBERT directly on downstream tasks to verify our hypothesis regarding the \[CLS\] token. Results in Table 2 are reported in the paper *HyperMixer: An MLP-based Low Cost Alternative to Transformers (Mai et al., 2023)*. The authors use models with 11M parameters for their experiments. As the authors stated in their paper, self-attention is considered the upper bound for their method.

---

### Official Review · Reviewer_ALq8 · 2025-10-31

**Soundness:** 2
**Presentation:** 4
**Contribution:** 2
**Rating:** 4
**Confidence:** 4

**Summary:**

This paper proposes replacing the per-layer attention weights in a transformer with weights from a learned hypernetwork that is shared across all layers. This makes the architecture more modular and reduces the number of parameters since the same hypernetwork is shared across all layers. The hypernetwork used in the experiments consists of a 4-head and 2-layer transformer decoder. Experiments and comparisons are carried out on a BERT model and the authors create their version of BERT (HyperBERT) that uses a shared hypernetwork in place of attention parameters. Experiments on GLUE show that HyperBERT can on average match BERT with 6% fewer parameters. In addition, the authors provide a theoretical analysis of the parameter efficiency.

**Strengths:**

* **Clarity**  The paper is generally well written and is easy to follow.
* **Results** The paper has good results for the range of the experiments presented (GLUE, BERT etc). In general the paper matches BERT with fewer parameters and with the hypernetwork.
* **Novelty** According to the authors, the paper is the first pre-training study to replace multi-head attention with MLPs generated by a shared hypernetwork. The problem of replacing MHA with a learned network is interesting in itself. [Disclaimer: I have not extensively read through the literature on hypernetworks, so I cannot verify if this is indeed the case.]
* **Ablations** There are detailed ablations for the different tasks, e.g retraining on MNLI to investigate the source of the performance gap etc.
* **Theoretical Analysis** The authors provide a theoretical analysis to support and ground the arguments in the paper.

**Weaknesses:**

* The paper and its results are all focused on Encoder-only style Transformer models (BERT). This raises the question of whether this would transfer to decoder-only style transformers which are more common for generative models etc. In this case how could one for example ensure causality in the learned operator ?
* What is the overall goal of the hypernetwork ? The decrease in parameters (a 6% decrease) does not look significant or helpful to me. Generally, I would expect that parameter sharing would lead to a good drop in parameter size, which could help reduce the memory requirements for training or fine-tuning e.g in LORA. Not too sure how far 6% might go! Is there a chance to get better gains in number of parameters as we scale?
* How does introducing a hypernetwork affect the runtime/memory usage of the model during training and inference ? Is this slower or is this faster ?
* The low rank assumption in the theory seems critical for a compact representation/hypernetwork, however the assumption is not well supported. Also why is r@90/L significantly smaller for T5-large (0.542) compared to the others (>0.8) ?
* One of the premises of the paper is that previous work is limited to small-scale experiments/tasks. One could argue that this is the same for this paper, which uses a small dataset wikitext101 and small network. How does this method scale ? Does it work beyond smaller tasks like GLUE and datasets like Wikitext ?

**Questions:**

See weaknesses

---

> ### Author Response · Authors · 2025-11-15
>
> We thank Reviewer ALq8 for their time and valuable feedback. We are pleased to address the concerns raised.
>
> ### **1. Transferring to Decoder-Only Models**
> The current HyperBERT design does not directly transfer to decoder-only transformers because our shared hypernetwork conditions on information that depends on the entire input sequence in its present form. When the same generator is invoked multiple times across layers, layer calls could access future tokens and violate causality even with a standard causal mask. With a shared hypernetwork, the conditioning itself must be causal.
>
> As part of ongoing work, we are pursuing two causal designs: (*i*) A shared hypernetwork that can provide causal MLPs for token mixing. Each token embedding is updated with information up to that token. (*ii*) A Mamba-style backbone model (*Gu & Dao, 2024*). One key modification of Mamba compared to previous SSMs is making the $A_t,B_t,C_t$ matrices from time-invariant to time-varying. In the Mamba model, the $A_t,B_t,C_t$ matrices depend on the input token with "hypernetworks" $s_B(x),s_C(x),s_\Delta(x)$. This architecture aligns with our generator-executor paradigm and preserves causality. We believe it is a promising route for future generative HyperBERT models.
>
> We will include a brief schematic and ablation plan in the appendix of the camera-ready version.
>
>
> ### **2. How Efficiency Improves with Depth and Sequence Length**
> Our method's advantages in both parameter and compute efficiency amplify as models grow in depth and context length.
>
> The first scaling axis is model depth. Our transferability experiment was designed to test exactly this. When we transplanted the same shared hypernetwork from the 4-layer model into a deeper 6-layer model, the parameter savings increased from 6% to 11%. This demonstrates that decoupling the parameter generation from the network's depth leads to efficiency gains as layers are added. This shows a clear path to more significant gains in larger-scale models.
>
> The second scaling axis is sequence length. Our theoretical FLOPs analysis in Appendix shows that HyperBERT's computational cost scales better with sequence length than standard BERT.
>
> To better illustrate how parameter efficiency and runtime scale with context length and model depth, we provide runtime and VRAM usage comparison results in our answer to Reviewer ALq8's third question below. The results address the point about memory requirements for training or fine-tuning. Furthermore, by fine-tuning the shared hypernetwork instead of layer-specific Q/K/V/O matrices, the fine-tuning parameter count becomes largely depth-independent. As shown in the tables below, HyperBERT enables larger batch sizes or longer sequences on the same hardware.
>
> The overall goal of the shared hypernetwork is to validate a more modular, transferable, and scalable architecture based on a shared hypernetwork. Instead of independent parameters for each layer, we centralize and externalize the model's sequence-understanding and weight-generation mechanism into a single shared hypernetwork. The 6% parameter reduction demonstrates viability at small depth. However, as shown in our paper and tables below, both runtime and memory advantages increase as model depth and context length grow.

---

> > ### Author Response · Authors · 2025-11-15
> >
> > ### **3. Empirical Runtime and VRAM Benchmark**
> > As described in our paper, introducing the shared hypernetwork adds a fixed hypernetwork cost. Thus, at short sequences, HyperBERT is slower than vanilla BERT. As sequence length grows, that fixed cost is amortized and HyperBERT becomes faster. On memory, HyperBERT consistently reduces peak VRAM usage during training and inference especially in the deeper models. To practically validate this, we benchmarked the BERT baseline and HyperBERT models across multiple configurations under the same setting on an RTX 4090 GPU, running 1000 iterations per configuration with batch size equal to 1. We report the average runtime per sample and the peak GPU memory usage. Full results are in the tables below.
> >
> > 4-layer models, $d_{model}= 768$, $h=128$:
> >
> > |Model|Mode|Sequence length|Avg ms/sample|VRAM Usage (GB)|
> > |-|-|-|-|-|
> > |BERT|Train|2048|20.96|2.31|
> > |HyperBERT|Train|2048|27.92|1.72|
> > |BERT|Train|3072|33.68|3.42|
> > |HyperBERT|Train|3072|28.49|2.26|
> > |BERT|Train|4096|49.22|4.76|
> > |HyperBERT|Train|4096|29.46|2.79|
> > |BERT|Train|6144|106.04|8.91|
> > |HyperBERT|Train|6144|40.37|4.42|
> >
> > |Model|Mode|Sequence length|Avg ms/sample|VRAM Usage (GB)|
> > |-|-|-|-|-|
> > |BERT|Inference|2048|4.76|0.88|
> > |HyperBERT|Inference|2048|7.88|0.82|
> > |BERT|Inference|3072|8.59|1.19|
> > |HyperBERT|Inference|3072|8.45|1.13|
> > |BERT|Inference|4096|13.07|1.49|
> > |HyperBERT|Inference|4096|9.08|1.43|
> > |BERT|Inference|6144|29.30|2.67|
> > |HyperBERT|Inference|6144|11.17|2.17|
> >
> > 8-layer models, $d_{model}=1024$, $h=128$:
> >
> > |Model|Mode|Sequence length|Avg ms/sample|VRAM Usage (GB)|
> > |-|-|-|-|-|
> > |BERT|Train|2048|43.50|4.24|
> > |HyperBERT|Train|2048|54.21|2.87|
> > |BERT|Train|3072|69.84|6.15|
> > |HyperBERT|Train|3072|55.63|3.59|
> > |BERT|Train|4096|101.59|8.48|
> > |HyperBERT|Train|4096|56.01|4.30|
> > |BERT|Train|6144|194.81|14.62|
> > |HyperBERT|Train|6144|62.27|5.75|
> >
> > |Model|Mode|Sequence length|Avg ms/sample|VRAM Usage (GB)|
> > |-|-|-|-|-|
> > |BERT|Inference|2048|9.18|1.32|
> > |HyperBERT|Inference|2048|14.34|1.14|
> > |BERT|Inference|3072|17.14|1.63|
> > |HyperBERT|Inference|3072|15.15|1.44|
> > |BERT|Inference|4096|26.83|1.96|
> > |HyperBERT|Inference|4096|16.02|1.75|
> > |BERT|Inference|6144|54.46|2.96|
> > |HyperBERT|Inference|6144|17.21|2.36|
> >
> > We will add these practical runtime and VRAM usage results to our paper.
> > ### **4. Low-Rank Assumption**
> > Our low-rank assumption is grounded in prior work that has empirically and theoretically demonstrated redundancy across attention heads and layers, as cited in our paper. Our empirical analysis in Table 5 was intended to further validate this for our theoretical framework. T5-large appears as a distinct outlier in Table 5. We noted that this applies to T5-small and T5-base as well, which also have lower $r@90/L$ values than other 6-layer and 12-layer models. We hypothesize that it is because of T5's encoder-decoder architecture and pre-training objectives while other models are either encoder-only or decoder-only architectures. This does not contradict our claim that low rank is a useful and broadly observed prior.
> >
> > To confirm our assumption is not dependent on these T5-based outliers, we re-ran the power law fit after removing T5-small/base/large models. The sublinear growth persists:
> >
> > |Threshold|$\alpha$|$\beta$|
> > |-|-|-|
> > |80|0.965|0.937|
> > |85|0.886|0.983|
> > |90|0.965|0.977|
> > Thus, the T5 family is a family-specific deviation rather than a counterexample to the trend we leverage.
> >
> > ### **5. Scope and Experiment Scale**
> > We agree that our experiments are modest compared to contemporary LLM pretraining. Our goal here is orthogonal: to test whether a single shared hypernetwork can replace all attention projections in an encoder-only model under a controlled compute budget. Within this scope, our setting is comparable to or larger than prior works replacing attention projections with hypernetwork mechanisms either implicitly or explicitly. (e.g., HyperMixer (*Mai et al., 2023*): ~11M without pre-training; HYLA (*Schug et al., 2025*): 10-50M models amplifying the hypernetwork mechanism of linear attentions.) To probe scalability, we provide three diagnostics: (*i*) Transferability experiments. When reusing the same hypernetwork from 4- to 6- layer models, the parameter savings increase from 6% to 11%. (*ii*) FLOPs analysis. We theoretically predict crossovers where HyperBERT becomes faster as context length grows. (*iii*) Wall-clock and VRAM usage tables with different configurations. We provide a practical runtime and VRAM usage report above. As shown in the tables, the speed and memory-efficiency gains in both training and inference increase as context length and model depth grow.
> >
> > We do not claim SOTA LLM scale. Instead, we present feasibility and scaling signals. When scaling up HyperBERT, the key consideration is expressivity. A fixed-capacity hypernetwork cannot support an infinitely deep model. However, our generator-executor paradigm makes this trade-off explicit and tunable, which we believe is a promising direction for scaling future architectures.

---

### Official Review · Reviewer_otMK · 2025-11-01

**Soundness:** 4
**Presentation:** 3
**Contribution:** 3
**Rating:** 8
**Confidence:** 4

**Summary:**

The authors proved that a shared hypernetwork can approximate multi-head self-attention with fewer parameters. They then built HyperBERT, a BERT variant where a single 2-layer Transformer decoder dynamically generates MLPs to replace all attention mechanisms across all layers. When pre-trained on WikiText-103, HyperBERT matched standard BERT's GLUE performance with 6% fewer parameters and showed better transfer learning to deeper models. This is the first work to successfully replace multi-head self-attention with hypernetwork-generated MLPs at the scale of full language model pre-training.

**Strengths:**

1. The paper is well-structured, with clear writing and clarity of thought
2. Discussion of related work is extensive, and motivates the ideas explored in the paper well
3. The theoretical / mathematical analysis is detailed and rigorous, and provides a strong motivation for the proposed architecture and empirical experiments
4. Empirical methodology is well-explained and rigorous
5. Empirical results are strong vs other approaches

**Weaknesses:**

1. The main model used for comparison, BERT, is six years old. It would be better if the authors benchmarked / based their method off more recent (and perhaps SotA) transformer models and architectures
2. The authors only test on small-scale models, relative to those which are commonly used today for language modelling, it would be interesting to see the impact on models of at least size ~1B parameters
3. The paper would benefit from a diagram illustrating their new architecture
4. The reduction in parameter count, 6%, is somewhat small.

**Questions:**

1. How do the authors think their method will be affected as you scale up the model size into the billions of parameters?
2. Do the authors have any intuition for how the performance in the encoder-only setting will transfer to the decoder-only setting? (i.e. GPT-style transformers rather than BERT-style)
3. Given the attention layer replacement becomes an MLP (with dynamically generated weights), could this also subsume whatever computations are happening in the fixed MLP (FFN) sublayers? I.e. could you use the hyper network to replace almost all the computations in the transformer, and not just attention?

---

> ### Author Response · Authors · 2025-11-15
>
> We thank Reviewer otMK for their time and highly positive and insightful review. We are pleased to address the weaknesses and the raised questions.
>
> ### **1. Baseline and Model Scale**
>
> We agree that BERT is an old baseline. We chose it because it is a standard, well-understood encoder-only architecture. This provides a controlled setting to prove the viability of replacing the multi-head self-attention mechanism with a shared hypernetwork, which is our paper's core contribution.
>
> Regarding the small scale, we agree that our experiments are modest compared to contemporary LLM pretraining. Our goal here is orthogonal: to test whether a single shared hypernetwork can replace all attention projections in an encoder-only model under a controlled compute budget. Within this scope, our setting is comparable to or larger than prior works replacing attention projections with hypernetwork mechanisms either implicitly or explicitly.
>
> ### **2. Scaling to Larger Models**
> Our method's advantages in both parameter and computational efficiency amplify as models grow in depth and context length.
>
> The first scaling axis is model depth. Our transferability experiment was designed to test this. When we transplanted the same shared hypernetwork from the 4-layer model into a deeper 6-layer model, the parameter savings increased from 6% to 11%. This demonstrates that decoupling the parameter generation from the network's depth leads to efficiency gains as layers are added. This shows a clear path to more significant gains in larger-scale models.
>
> The second scaling axis is sequence length. Our theoretical FLOPs analysis in the Appendix shows that HyperBERT's computational cost scales better with sequence length than standard BERT.
>
> To prove this practically, we ran new benchmarks comparing the runtime and VRAM usage of the BERT baseline and HyperBERT under the same setting on an RTX 4090 GPU, running 1000 iterations per configuration with batch size equal to 1. We report the average runtime per sample and the peak GPU memory usage. Full results are in the tables below.
>
> 4-layer models, $d_{model}= 768$, $h=128$:
>
> |Model|Mode|Sequence length|Avg ms/sample |VRAM Usage (GB)|
> |-|-|-|-|-|
> |BERT|Train|2048|20.96|2.31|
> |HyperBERT|Train|2048|27.92|1.72|
> |BERT|Train|3072|33.68|3.42|
> |HyperBERT|Train|3072|28.49|2.26|
> |BERT|Train|4096|49.22|4.76|
> |HyperBERT|Train|4096|29.46|2.79|
> |BERT|Train|6144|106.04|8.91|
> |HyperBERT|Train|6144|40.37|4.42|
>
> |Model|Mode|Sequence length|Avg ms/sample|VRAM Usage (GB)|
> |-|-|-|-|-|
> |BERT|Inference|2048|4.76|0.88|
> |HyperBERT|Inference|2048|7.88|0.82|
> |BERT|Inference|3072|8.59|1.19|
> |HyperBERT|Inference|3072|8.45|1.13|
> |BERT|Inference|4096|13.07|1.49|
> |HyperBERT|Inference|4096|9.08|1.43|
> |BERT|Inference|6144|29.30|2.67|
> |HyperBERT|Inference|6144|11.17|2.17|
>
> 8-layer models, $d_{model}=1024$, $h=128$:
>
> | Model | Mode | Sequence length | Avg ms/sample | VRAM Usage (GB)|
> |-|-|-|-|-|
> |BERT|Train|2048|43.50|4.24|
> |HyperBERT|Train|2048|54.21|2.87|
> |BERT|Train|3072|69.84|6.15|
> |HyperBERT|Train|3072|55.63|3.59|
> |BERT|Train|4096|101.59|8.48|
> |HyperBERT|Train|4096|56.01|4.30|
> |BERT|Train|6144|194.81|14.62|
> |HyperBERT|Train|6144|62.27|5.75|
>
> | Model | Mode | Sequence length | Avg ms/sample | VRAM Usage (GB)|
> |-|-|-|-|-|
> |BERT|Inference|2048|9.18|1.32|
> |HyperBERT|Inference|2048|14.34|1.14|
> |BERT|Inference|3072|17.14|1.63|
> |HyperBERT|Inference|3072|15.15|1.44|
> |BERT|Inference|4096|26.83|1.96|
> |HyperBERT|Inference|4096|16.02|1.75|
> |BERT|Inference|6144|54.46|2.96|
> |HyperBERT|Inference|6144|17.21|2.36|
>
> Introducing the shared hypernetwork adds a fixed hypernetwork cost. Thus, at short sequences, HyperBERT is slower than vanilla BERT. As sequence length grows, that fixed cost is amortized, and HyperBERT becomes faster. As shown in the tables, the runtime and VRAM advantages of HyperBERT become significantly larger as context length and model depth increase. This directly enables training on longer contexts or with larger batch sizes on the same hardware.
>
> When scaling up HyperBERT, the key consideration is expressivity. A fixed-capacity hypernetwork cannot support an infinitely deep model. Scaling up HyperBERT requires determining the optimal proportion of total parameters to allocate to the hypernetwork. However, our generator-executor paradigm makes this trade-off explicit and tunable, which we believe is a promising direction for scaling future architectures.
>
> We will add these practical runtime and VRAM usage results to our paper.
>
> ### **3. Additional Diagram**
>
> We thank Reviewer otMK for this suggestion. In the final version, we will revise Figure 1 to more clearly detail the architectural components and the flow of information for weight generation.

---

> ### Author Response · Authors · 2025-11-15
>
> ### **4. Transferring to Decoder-only Models**
>
> The current HyperBERT design does not directly transfer to decoder-only Transformers, as we note in our Limitations section. The problem is that our shared hypernetwork conditions on the entire input sequence. When the same generator is invoked in later layers, it could access future token information.
>
> However, this implicit hypernetwork mechanism is inherent to both encoder-only and decoder-only architectures. Mamba (*Gu & Dao, 2024*) also includes this implicit hypernetwork mechanism to improve traditional SSM models (by making the $B_t, C_t, \Delta_t$ matrices dependent on the input token $x_t$) and achieves state-of-the-art performance. This architecture aligns with our generator-executor paradigm and keeps causality. We believe the hypernetwork mechanism is critical for language modeling. We are actively exploring efficient methods for transferring HyperBERT to modern language models. The experience with HyperBERT is valuable.
>
> ### **5. Replacing FFN Sublayers**
>
> This is an excellent and insightful question. The answer is yes. This is a key and exciting extension of our generator-executor paradigm.
>
> Theoretically, there is nothing stopping the hypernetwork from generating the weights for FFNs as well. We are actively exploring this, and our preliminary results are promising. We find that a single, shared hypernetwork can indeed generate parameters to replace both the multi-head self-attention and the subsequent FFN, unifying the entire Transformer block into a dynamically generated operator.
>
> This design enhances modularity and unlocks more flexible architectures such as adaptive depth. The shared hypernetwork could learn to dynamically generate computational blocks based on context and decide whether to continue processing or to exit early.
>
> We thank Reviewer otMK again for their thoughtful feedback. We would be pleased to discuss this direction further.

---

### Meta-Review · Area_Chair_yx7N · 2025-12-28

**Summary:**

The authors propose a method to replace the attention MLPs in a Bert Transformer Encoder with an MLP with weights produced by a hypernetwork from context, allowing to share trainable weights over layers. The paper was received with negatively leaning scores of (0,4,4,8) pre-discussion, where the reviewers raised concerns regarding the chosen architecture, the limited scope of experiments (small scale), the novelty of contribution, and the significance of gains achieved by the modified architecture (see below).

After looking at the reviews, the paper, and the author answers, I believe that the negatively leaning reviewers (4,4) would most likely not have increased their scores, as I think their concerns are not sufficiently addressed by the authors answers. Meanwhile, the 0-score reviewer has a very strong opinion on rejecting this work. Even when partially disregarding the 0-score review (due to an extreme stance that has been partially rebutted by the authors), I believe this paper is not ready for publication at ICLR. Mainly, concerns regarding chosen architecture, scale of experiments and significance of results are not living up to the original motivation of the work. Therefore, I recommend to reject the paper.

**Reviewer Concerns:**

*1) Outdated architecture choice of BERT.* The authors acknowledge this limitation in the original paper and discussion but it remains an issue that the results might not easily translate to current-gen architectures.

*2) Experiments too small-scale.* The authors acknowledge this limitation but don't remedy it. Since the original motivation for this work is better scaling in large-scale models, this remains an issue.

*3) Limited Novelty.* The authors successfully rebut that there is not novelty at all. However, it remains the case that novelty in architecture are incremental and straight-forwardly assembled from existing hypernetwork literature. This does not have to be an issue in itself but shows that conceptual novelty is not a particular strength of this work.

*4) Significance of results.* The authors correctly pointed out that the gains improve for large-scale and provide further evidence in their answers. While this concern is partially addressed, I believe it still remains an issue that this is not shown on larger scale experiments if it is such a crucial aspect of this work.

**Reviewer Scores:**

I believe the scores would have remained as they are. Reviewer otMK was already positive and would have remained with 8. Reviewers ALq8 and mtKV concerns are only partially addressed, probably not convincingly enough to make them increase their scores. Reviewer NJeA with score 0 has a very strong opinion that this paper should be rejected, which the authors did not manage to change.

---

### Decision · Program_Chairs · 2026-01-26

Reject